



# The Effect of Soil Moisture Anomalies on Maize Yield in Germany

Michael Peichl, Stephan Thober, Volker Meyer, and Luis Samaniego

UFZ - Helmholtz Centre for Environmental Research, Leipzig, Germany

*Correspondence to:* Michael Peichl (Michael.Peichl@ufz.de)

**Abstract.** Crop models routinely use meteorological variations to estimate crop yield. Soil moisture, however, is the primary source of water for plant growth. The aim of this study is to investigate the intra-seasonal predictability of soil moisture to estimate silage maize yield in Germany. It is also evaluated how approaches considering soil moisture perform compared to those using only meteorological variables. Silage maize is one of the most widely cultivated crops in Germany because it is

used as a main biomass supplier for energy production in the course of the German Energy Transition. Reduced form fixed effect panel models are employed to investigate the relationships in this study. These models are estimated for each month of the growing season to gain insights into the time varying effects of soil moisture and meteorological variables. Temperature, precipitation, and potential evapotranspiration are used as meteorological variables. Soil moisture is transformed into anomalies which provide a measure for the inter-annual variation within each month. The main result of this study is that soil moisture

anomalies have predictive skills which vary in magnitude and direction depending on the month. For instance, dry soil moisture anomalies in August and September reduce silage maize yield more than 10 % other factors being equal. On the contrary, dry anomalies in May increase crop yield up to 7 % because absolute soil water content is higher in May compared to August due to its seasonality. With respect to the meteorological terms, models using both temperature and precipitation have higher predictabilities than models using only one meteorological variable. Also, models employing only temperature exhibit elevated

effects.

## 1 Introduction

In the course of the German Energy Transition, the demand for biomass has increased considerably with silage maize being an important plant for high dry matter yields. The share of the total production in agriculture was 18 % in 2014 (Die Landwirtschaft Band 1, 2014), with an increasing share of agricultural area used for silage maize from 15.4 % in 2010 to 17.7 % in 2015

(Statisitisches Bundesamt, 2011, 2016). With that in mind, the observed susceptibility of silage maize towards extreme dry conditions during summer time supports the detection of relevant factors for yield (as for instance in 2015, Becker et al., 2015; Bundesministerium für Ernäherung und Landwirtschaft, 2015). Knowing the determinants of maize variation can help to mitigate welfare losses. For instance, detrimental effects of soil moisture shortage and abundance can be mitigated by the means of irrigation and drainage and thus are key for targeted and efficient development of adaptation measures (Chmielewski,

25 2011).





In general, two different kinds of modeling approaches are employed to assess the impact of weather or climate on the agricultural sector. These are structural (integrated assessment) models and reduced form models (Auffhammer and Schlenker, 2014). Whilst structural approaches specify the economic behavior based on theoretical models and assumptions and thus have "the ability to make predictions about counterfactual outcomes and welfare" (Chetty, 2009), the advantage of reduced form approaches is "transparent and credible identification" (Chetty, 2009) by exploiting the exogenous variation of key parameters (Timmins and Schlenker, 2009). Regression models are used to estimate the variation in the dependent variable within various sectors by the means of *damage* or *dose-response functions* (Hsiang, 2016; Carleton and Hsiang, 2016). In the agricultural sector, the major explanatory variables are temperature based (Carleton and Hsiang, 2016; Lobell et al., 2008, 2011b; Schlenker et al., 2005; Schlenker and Lobell, 2010). The use of temperature as the main explanatory variable is questioned in this study by using reduced form models to identify the impact of different determinants on crop yield.

In the agricultural context, most advances have been made regarding dose-response functions through the development of temperature estimates on high spatial and temporal resolutions (Hsiang, 2016). Based on this data, many studies employ a precise term which integrates cumulative exposure to specific temperature ranges over the growing period as major explanatory variable. Those are defined as growing degree days (Schlenker et al., 2006; Deschenes and Greenstone, 2007) and accumulated measures of extreme heat above a certain threshold, as for instance extreme, heat, killing, or damage degree days (Annan and Schlenker, 2015; Burke and Emerick, 2016; Butler and Huybers, 2013, 2015; Lobell et al., 2011a, 2013; Ortiz-Bobea and Just, 2013; Roberts et al., 2013; Urban et al., 2012, 2015a; Schlenker and Roberts, 2006, 2009; Schlenker et al., 2013; Teixeira et al., 2013). Schlenker and Roberts (2009) showed, that the time a plant is exposed to temperature above a threshold is able to explain almost half of its variation in yields. For corn, this threshold is estimated to be 29 °Celsius. Thus, it is highly recommended to account for nonlinearity. Extreme degrees days (EDD) are considered as the best predictor of crop yield variation (Auffhammer and Schlenker, 2014; Carleton and Hsiang, 2016).

Recent research suggests, that the main reason of the importance of EDD is the high correlation with measures of cumulative evaporative demand (Urban et al., 2015a), as for instance vapor pressure deficit (VPD, Roberts et al., 2013; Lobell et al., 2013). There is evidence, that the effect of EDD and measures for evapotranspirative demand is overstated when neglecting proper control for water supply (Ortiz-Bobea, 2013; Basso and Ritchie, 2014). For instance, soil moisture is considered a major limiting factor to maize growth (Andresen et al., 2001). Extreme high temperature amplifies the impact of soil moisture deficit because of surface-atmosphere coupling (Mueller and Seneviratne, 2012), but the opposite is not necessarily the case as droughts occur independently of heat (Basso and Ritchie, 2014). Urban et al. (2015b) highlight the impact of interactive effects between VPD and water supply to further improve model predictability. In Germany, a recent statistical impact assessment of weather fluctuations affecting maize and winter wheat recognizes water shortage as major limiting factor (Gornott and Wechsung, 2015, 2016; Conradt et al., 2016). These studies employ proxies to control for the primary source of water, such as precipitation and measures for evapotranspirative demand. The water holding capacity of the soil and the persistence of soil moisture is often not considered.

One basic assumption in EDD is that temperature effects are additive substitutable, which means that their impact is constant for all development stages of the plant. This assumption is rejected in both agronomic studies (de Bruyn and de Jager, 1978;



Sinclair and Seligman, 1996; Tubiello et al., 2007; Wahid et al., 2007) and large-scale empirical analyses (Lobell et al., 2011a; Ortiz-Bobea, 2011; Ortiz-Bobea and Just, 2013; Berry et al., 2014). These show, that the susceptibility to high temperatures is elevated during flowering (i.e. tasseling, silkening, and pollination). Similar to heat measurements, the sensitivity to water stress is dependent on the development stage of the plant (FAO Water, 2016). For instance, it is shown for climate projections in India

that a more uneven distribution of precipitation within a season overturns positive effects of an increase in total precipitation (Fishman, 2016). It is argued to control for intra-seasonal varying weather induced effects on crop yield variation. This issue is amplified for precipitation controls compared to temperature. The distribution of measures such as EDD partially overlaps with the sensitive phase of plant growth (see Figure A14 of Schlenker and Roberts, 2009), but precipitation, as control for water supply, is commonly aggregated for the entire growing season (Annan and Schlenker, 2015; Burke and Emerick, 2016; Roberts

et al., 2013; Schlenker and Roberts, 2006, 2009, among others). These studies do not explicitly account for seasonality of water supply related effects. Overall, controls for meteorological effects averaged over the entire season may bias the estimated dose-response function and diminish the predictive power of the models, because they do not account for the seasonal interaction between water supply and water demand (Urban et al., 2015b).

Based on this analysis, it is the main aim of this study to investigate the intra-seasonal predictability of soil moisture to

estimate silage maize yield in Germany. It is also evaluated how approaches considering soil moisture perform compared to those using meteorological variables. The examined hypothesis are, that a) models with soil moisture are better able to predict yield than meteorology-only approaches and that b) temporal patterns in the seasonal effects of the explanatory variables matter, i.e. there is no additive substitutability. In order to analyze these hypotheses, the intra-seasonal effects of soil moisture and meteorological variables for non-irrigated arable land in Germany are examined in this study. In detail, the following research

questions are addressed: 1) Is there predictability of soil moisture additionally to meteorology? 2) If so, how does it compare to the one by meteorological determinants? 3) Is there temporal pattern in the seasonal effects of all explanatory variables (meteorology and soil moisture)? Along this analysis we also evaluate 4) how models based on different meteorological determinants perform compared to each other.

To answer this research questions, a reduced form panel approach is employed to examine the non-linear intra-seasonal

partial effects of soil moisture anomalies and the meteorological variables temperature, potential evapotranspiration, and precipitation. For this purpose, we use a new data set which is additionally comprised of soil moisture anomaly data. The aim is to evaluate whether soil moisture anomalies have predictive skills and how the effects differ from those using only meteorological variables. Soil moisture and any derived index is highly autocorrelated in time and thus provide an integrated signal of the meteorological conditions in the preceding and subsequent months (e.g., Orth and Seneviratne, 2012; Samaniego et al., 2013).

This persistence does not allow for cumulative measures as those used for temperature, but it avoids the inflation of the error terms. Commonly, the predictive power of models only employing meteorological variables can be improved by accounting for the regional specific temporal distribution of the phenological stages (Dixon et al., 1994). The integrated signal of the meteorological conditions provided by any measure derived from soil moisture, however, allows the employment of monthly averages to account for these intra-seasonal effects. In our study, it is implicitly controlled for the interaction of both variables

controlling for water supply and water demand, because these show high correlation on a monthly basis. Different model





configurations for each month of the growing season are compared by model selection criteria to allow conclusions about the effect of soil moisture anomalies on the explanatory power of the model and to test the assumption of additive substitutability. Further, the difference in explanatory power of models either using potential evapotranspiration or average temperature is evaluated. The partial effects of all covariates of the best model for each month are examined. For the purpose of a comprehensive examination, we also investigate the effects of wet anomalies.

## 2 Data

### 2.1 Yield Data

Annual yield data for silage maize are provided by the Federal Statistical Office of Germany for the administrative districts (rural districts, district-free towns, and urban districts) since the year 1999 (Statistische Ämter des Bundes und der Länder, 2017). The yield data are de-trended using linear regression for the period 1999 to 2015 to control for technical progress. A log transformation is applied to yield to better satisfy the normality assumption. This transformation also mitigates issues related to heteroscedasdicity and the estimates are less sensitive to outliers. All administrative districts with less than nine observations are removed from the analysis, because the influence of single observations points is too strong in these cases. The threshold nine has been chosen after exploring Cook's distance and evaluating the systematic omission of yield data by the administrative districts (Cook, 1977, 1979).

### 2.2 Soil Moisture Anomalies and Meteorology

The explanatory variables used in the study are the observed meteorological variables precipitation (P), average temperature (T), and potential evapotranspiration (E), as well as model-derived soil moisture. The mesoscale Hydrologic Model (mHM) has been used to estimate the soil moisture (Samaniego et al., 2010; Kumar et al., 2013). The model uses grid cells as primary unit and it accounts for various hydrological processes such as infiltration, percolation, evapotranspiration, snow accumulation, groundwater recharge and storage as well as fast and slow runoff. The parametrization process of the model is based on physical characteristic, as for instance soil texture. Three different forms of land cover are also integrated in the model, which are based on the CORINE Land Cover maps of 2006 (European Environmental Agency, 2009). However, no endogenous processes of land use management, as for instance drainage or irrigation, are considered within the model. The depth of the soil in each grid depends on the soil type used in mHM. Details can be found in Zink et al. (2017).

Soil moisture is further transformed into a soil moisture index (SMI), which is a non-parametric cumulative distribution function (cdf) derived from the absolute soil moisture estimated by mHM. A non-parametric kernel smoother algorithm has been used for the calculation of the cdf for each calendar month in accordance to the proposed method by Samaniego et al. (2013). It ranges from zero to one and represents an anomaly with respect to the monthly long term median in soil water (SMI $= 0.5$). Low values represent extreme dry soils and high values extreme wet ones. The SMI is calculated for entire Germany at a spatial resolution of 4 km. Monthly values of soil moisture are transformed to SMI for the period from 1951 to 2015. These





values have also been used for drought reconstruction (Samaniego et al., 2013). A similar procedure has been applied for the seasonal forecasts of agricultural droughts (Thober et al., 2015).

The monthly SMI values are categorized into seven classes which follow the notion of the US drought monitor and the German Drought Monitor (Zink et al., 2016). This stepwise approach allows to measure nonlinear effects of soil moisture.

The dry categories SMI $\leq 0.1$, $0.1 <$ SMI $\leq 0.2$, and $0.2 <$ SMI $\leq 0.3$ are denoted as severe drought, moderate drought and abnormally dry, respectively. The wet quantile intervals between $0.7 <$ SMI $\leq 0.8$, $0.8 <$ SMI $\leq 0.9$, and $0.9 <$ SMI are labeled as abnormally wet, abundantly wet and severely wet, respectively. The interval between $0.3 <$ SMI $\leq 0.7$ serves as reference and characterizes normal situations. In the rest of this , the terms soil moisture anomalies and soil moisture index (SMI) are used synonymously because of this categorization.

Daily data of precipitation and temperature are obtained from a station network operated by the German Weather Service (Deutscher Wetterdienst, 2017). Details on interpolation can be found in Zink et al. (2017). These daily values are also used to force mHM. For the analysis in this study, all daily values are aggregated to monthly ones conserving the mass and energy of the daily observations.

Further, we introduce Potential Evapotranspiration (E) as a measure for evaporative demand. E is calculated by the equation
of Hargreaves and Samani (1985) based upon extraterrestrial radiation and temperature and is measured in water evaporation (mm d$^{-1}$):

$$E = \kappa R \sqrt{T_\delta}(T + 17.8), \tag{1}$$

where $\kappa$ is a free parameter ($^\circ$C$^{-1.5}$) that compensates for advection of water vapor (mm d$^{-1}$), R is extraterrestrial radiation converted into equivalent water evaporation, and $T_\delta$ is the temperature difference between daily maximum and daily minimum
temperature ($^\circ$C). The term $T+17.8$ is an approximation of saturated vapour pressure, whereas the term $T_\delta$ is an approximation of cloudiness. 17.8 is an empirical constant found by calibration.

More complex alternatives exist, as for instance the standard method of United Nations Food and Agriculture Organization after Penman and Monteith (Monteith, 1981). These data for example use net radiation that is more difficult to estimate at the national scale in comparison to temperature particularly due to the lack of consistent observations. Similar to Vapor
Pressure Deficit, which has been introduced as effective crop yield predictor (Roberts et al., 2013; Lobell, 2013), potential evapotranspiration has a more direct physical link to crop water requirements than temperature. One goal of this study is to evaluate whether potential evapotranspiration provides improved yield estimates in comparison to temperature.

All meteorological variables are normalized to ease the comparison among different months. After this transformation, the variables have a mean of zero and a standard deviation of one. The original mean and standard deviation of the meteorological
variables are depicted in Table 1 for completeness.



**Table 1.** Mean and Standard Deviation of the Meteorological Variables, averaged over Germany. Data are obtained by the Germany Weather Service.

|  | May | | June | | July | | August | | September | | October | |
|---|---|---|---|---|---|---|---|---|---|---|---|---|
|  | Mean | SD | Mean | SD | Mean | SD | Mean | SD | Mean | SD | Mean | SD |
| P (monthly sum in mm) | 75.74 | 39.84 | 69.71 | 33.15 | 89.48 | 39.72 | 84.04 | 43.68 | 63.88 | 32.62 | 57.72 | 27.28 |
| T (monthly average in °C) | 13.46 | 1.42 | 16.52 | 1.45 | 18.48 | 1.74 | 17.90 | 1.57 | 14.07 | 1.63 | 9.64 | 1.83 |
| E (monthly average in mm) | 115.23 | 12.15 | 133.42 | 12.21 | 139.10 | 16.52 | 115.24 | 13.55 | 70.33 | 8.73 | 36.82 | 4.69 |

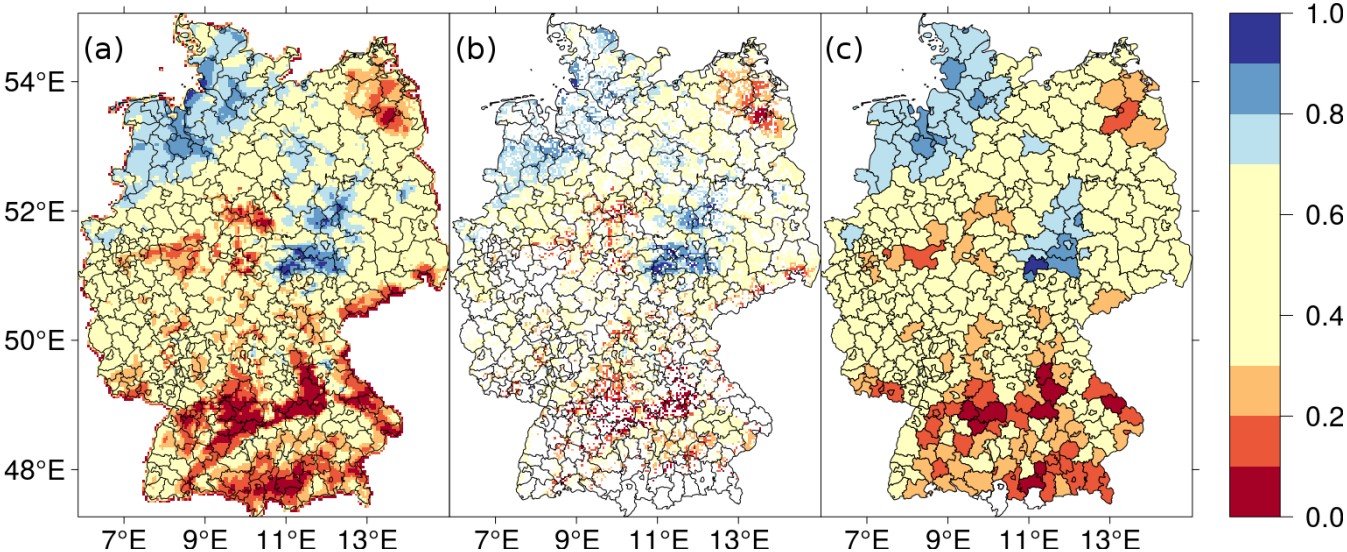

**Figure 1.** Illustration of the spatial processing of the SMI data of May 2003. On the left side, one can see the SMI with the $4 \times 4$ km$^2$ grids. In the middle, the data are masked with the $0.1 \times 0.1$ km$^2$ *non-irrigated arable land*-class of the CORINE Land Cover. Those data are than averaged over all the grid cells which are inside an administrative district. This is done for each district and the map on the right is derived. The processing steps shown in panel (a) and (b) are shown here exemplary for the soil moisture index for consistency, but these processing steps are applied to soil moisture fractions.

## 2.3 Spatial Processing

The explanatory variables (meteorology and soil moisture) are mapped onto the level of administrative districts to align with the spatial scale of the yield data. Maps st the different processing steps are shown in Fig. 1. Figure 1a depicts the $4 \times 4$ km$^2$ grid. These absolute soil moisture fractions are masked for *non-irrigated arable land*-class of the CORINE Land Cover (2006) at a $0.1 \times 0.1$ km$^2$ resolution to account for the variability due to heterogeneous land use within the administrative districts (Fig. 1b). The 0.1 km values are then averaged for each of the administrative district to obtain district level values (Fig. 1c).



Blank administrative districts occur because of the absent of *non-irrigated arable land* grid cells. These processing steps are also applied to the meteorological variables (P, T, E). The soil moisture fractions of each administrative district is then transformed into a percentile index (SMI) using the kernel density estimator explained above (Samaniego et al., 2013; Thober et al., 2015; Zink et al., 2016). An index reduces the probability of measurement errors and the estimated coefficients in the
regression models are supposed to be less prone to attenuation bias (Fisher et al., 2012; Auffhammer and Schlenker, 2014; Hsiang, 2016).

## 3   Regression Analysis

The main aim of this study is the identification of the monthly effects of soil moisture anomalies on crop yield. The model relates silage maize yield deviation (Y) to a stepwise function of soil moisture anomalies (SMI) and polynomials of the me-
teorological variables (P, T, E). Also, an error term is included which is composed of the fixed effects (c), a time-invariant categorical administrative district identifier, and the observation-specific zero-mean random-error term, which is allowed to vary over time ($\epsilon$). Each monthly model can be written as:

$$Y_{ik} = \sum_{j=1}^{6} \alpha_j \mathrm{I}(SMI_{ikm} \in \mathrm{C_j}) \tag{2}$$

$$+ \sum_{j=1}^{3} \beta_j (\mathrm{P}_{ikm})^j + \sum_{j=1}^{3} \gamma_j (\mathrm{T}_{ikm})^j + \sum_{j=1}^{3} \delta_j (\mathrm{E}_{ikm})^j \tag{3}$$

$$+ c_{im} + \epsilon_{ikm}. \tag{4}$$

The index $i$ represents the administrative districts, $k$ the years, and $m$ each month of the growing season, while the superscript $j$ refers to degrees of the polynomials. $\mathrm{I}(\cdot)$ is the indicator function of the soil moisture categories $\mathrm{C_j}$, being 1 if the SMI belong to class $j$ and 0 otherwise. The six classes are defined as severe drought (SMI $\leq 0.1$), moderate drought ($0.1 <$ SMI $\leq 0.2$), abnormally dry ($0.2 <$ SMI $\leq 0.3$), abnormally wet ($0.7 <$ SMI $\leq 0.8$), abundantly wet ($0.8 <$ SMI $\leq 0.9$) and severely wet
($0.9 <$ SMI), respectively. The estimated coefficients of the model are $\alpha$, $\beta$, $\gamma$, and $\delta$ and are constrained to be the same for all administrative districts.

The explanatory variables are correlated to each other (Table 2). Thus, higher non-orthogonal polynomials induce singularity in the moment matrix which cannot be inverted as required by the ordinary least-squares estimation of the coefficient. The polynomials are limited to degree three to avoid this and other detrimental consequences of multicollinearity such as the
inflation of the standard errors. Additionally, E and T are treated as mutually exclusive because of the high correlation of E and T (Table 2). The coefficients $\gamma$ or $\delta$ are set to 0, accordingly.

In addition to soil moisture, a meteorological and the fixed effect term are included to reduce omitted variable bias. The fixed effects account for the time-invariant confounding factors specific to each spatial unit as for instance average weather conditions. Thus, the coefficients of the exogenous variables are identified based on year-to-year variations and the analysis



**Table 2.** Comparison of Pearson Correlation Coefficients of the Exogenous Variables.

|         | May   | June  | July  | August | September | October | Average | Avg. June to Aug. |
|---------|-------|-------|-------|--------|-----------|---------|---------|-------------------|
| E / T   | 0.84  | 0.86  | 0.92  | 0.84   | 0.65      | 0.4     | 0.75    | 0.87              |
| E / P   | −0.38 | −0.38 | −0.52 | −0.52  | −0.56     | −0.15   | 0.42    | 0.47              |
| P / T   | −0.31 | −0.22 | −0.54 | −0.47  | −0.47     | −0.06   | 0.35    | 0.41              |
| SMI / E | −0.27 | −0.28 | −0.44 | −0.49  | −0.46     | −0.02   | 0.33    | 0.40              |
| SMI / P | 0.19  | 0.31  | 0.43  | 0.43   | 0.5       | 0.09    | 0.33    | 0.39              |
| SMI / T | −0.04 | −0.16 | −0.35 | −0.35  | −0.27     | 0.13    | 0.22    | 0.29              |

Annotation: Absolute values of the Pearson Correlation Coefficients are employed to calculated the averages
presented in the last two columns.

in this study can be considered as natural experiment (Auffhammer and Schlenker, 2014; Schlenker and Roberts, 2009). This interpretation is particularly suitable for SMI because this index, which describes deviations from them median, is per definition an anomaly. The meteorological term is included to account for important time-variant weather related factors. Endogenous variables are not included because these are considered as bad control in frameworks as those defined by Angrist and Pischke

(2008). For instance, prices are affected by weather realizations and climate and are thus defined as endogenous (Hsiang et al., 2013; Hsiang, 2016; Gornott and Wechsung, 2015, 2016).

Other studies additionally use annual fixed effects and interaction terms of both time and entity specific fixed effects to control for time specific confounding factors (e.g., Moore and Lobell, 2014). These terms are not used in this study because annual variation should be explicitly accounted for by the weather variation of the exogenous variables. Annual fixed effects would

diminish the entity specific inter-annual variation of the exogenous variables and thereby potentially amplify measurement errors (Fisher et al., 2012).

Various estimation approaches are used to evaluate the quality of the models. Models can be distinguished by the explanatory variables they use and the degree of polynomials in the meteorological terms. The maximum number of parameters estimated in a model is 12, excluding those of the fixed effects. The Bayesian Information Criteria (BIC) is used for model selection in

the next section. The BIC is composed of the maximum of the likelihood function for a particular set of variables as well as a penalty term (Schwarz, 1978). The latter adjusts the model selection criterion for the number of parameters to account for over-fitting. This allows to choose across models with different number of variables. The BIC criterion imposes a higher penalty on over-fitting compared to other model selection criteria based on maximum likelihood such as the Akaike Information Criterion (Akaike, 1973). The penalty particularly affects the soil moisture anomaly term because it always adds six parameters. Overall,

the model with the lowest BIC is preferred. To derive the BIC, a generalized linear model is fitted using the *glm* function (R Core Team, 2015).

Additionally, the models are evaluated according to their adjusted coefficient of determination (adj. $R^2$, Section 4.2). Ordinary least squares using the *lm* function (R Core Team, 2015) are employed with a dummy variable for each administrative




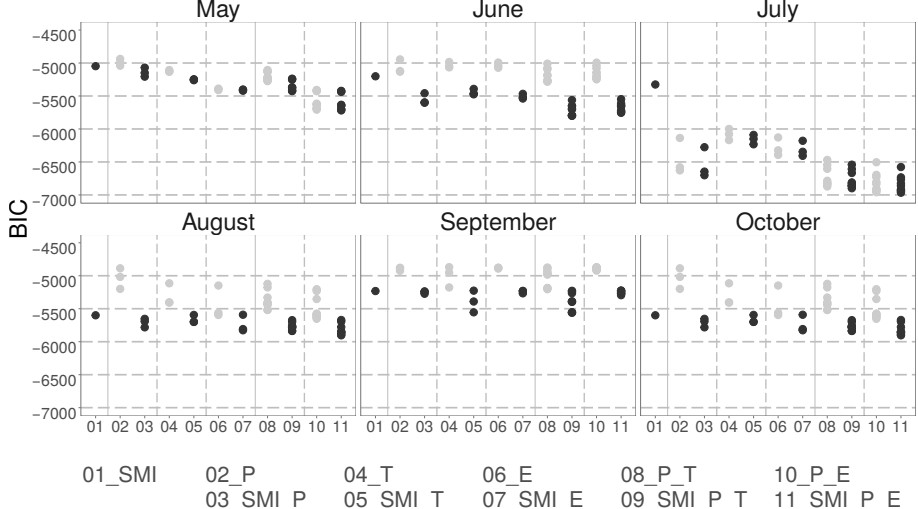

**Figure 2.** Each panel shows the BIC distribution of one month. Within the panels various models are compared, whilst the lowest marker is preferred. Each column represents a particular selection of variables. The markers represent different degrees of the polynomials in the meteorological term. The gray markers denote those models that neglect the SMI, whilst the black include it.

districts to explicitly account for the fixed effects. As default, a demeaning framework (Croissant and Millo, 2008) has been applied to investigate the model performance in terms of $R^2$. The estimated coefficients are the same for the least squares dummy variable regression, a demeaning framework, and maximum likelihood (BIC). This is in accordance to theory that normal distributed error terms estimators based on maximum likelihood and least squares are the same.

5      The standard errors of the coefficients are corrected for spatial autocorrelation. For this purpose, the Robust Covariance Matrix Estimator proposed by Driscoll and Kraay (1998) is employed to construct standard errors based on asymptotic formulas. No weights capturing decaying effects in space are used because the administrative districts have different areas and the spatial extent of SMI occurrences is heterogeneous. This can be regarded as comparable to block-bootstrapping on country-level, which has been used in many comparable studies relying on re-sampling methods (e.g. Butler and Huybers, 2015; Moore and

10 Lobell, 2014, 2015; Urban et al., 2015a, b). Further, serial correlation and heteroskedasdicity is also controlled for (White, 1980; Arellano, 1987). Overall, this approach is rather conservative but in alignment with the proposal of Angrist and Pischke (2008) to take the largest robust standard error as measure of precision.





## 4   Results and Discussion

### 4.1   Qualitative evaluation of different model configurations within the growing season

In this section, the Bayesian Information Criterion (BIC) is applied to evaluate the best combination with respect to soil moisture, meteorological variables, and the polynomial degrees of the latter. The BIC is calculated separately for each month
to assess the intra-seasonal variability.

The distribution of the BIC for the various model configurations is presented in Fig. 2., which shows one panel for each month of the growing season. Within the panels, models with different variable combinations in the meteorological term are separated by vertical lines. A model configuration is defined by a set of meteorological variables, the polynomial degree of each variable, and the stepwise function of the soil moisture anomalies. The complexity of the configurations increase stepwise
from the left to right within each panel. The model employing SMI as single explanatory variable is represented by a point on the left in each panel. The black markers indicate the models with soil moisture and gray markers without. The models 02 - 07 employ one meteorological variable each. These have three markers for the different degrees of the polynomials. The models 08 - 11 entail two meteorological variables and thus have nine markers.

The explanatory power is different across the months as indicated by the lowest marker within each panel. Overall, July has
the highest explanatory power. Nonlinear meteorological terms improve the fit of the model on the data in all model configurations (not shown). The preferred polynomial in the meteorological term is of degree three. The only exception is June, where the best model employs a second degree polynomial for P. Those observations are evident from an agronomic perspective, as for instance already early research employs curvilinear relationships between maize yield and meteorological variables (Thompson, 1969). Nonlinear configurations have been neglected so far in comparable approaches employing constant elasticity models in
Germany (Gornott and Wechsung, 2015, 2016; Conradt et al., 2016).

The composition of the meteorological term is evaluated by comparing the gray markers in Fig. 2. It is possible to asses the impact on the model fit of the single variables P, T, and E by the comparison of the configurations 02, 04, and 06, respectively. In May, most of yield variation is explained by E. In June and July, P contributes to model fit the most. In July, for instance, the explanatory power of a nonlinear P term is almost as good as the best combined configuration. September and October
are determined by T. However, in most months, using more than one meteorological variable results in the highest explanatory power. The only exception is October, where model 05 (SMI & T) exhibits the lowest BIC.

The difference in BIC between configuration 08 (P & T) and 10 (P & E) is small from June to August. This result can be expected because T and E are highly correlated in our sample (Table A2). The models with mixed meteorological terms in July and August slightly prefer E, while in June it is T. In the other months, the difference between T and E is comparatively
larger. In May, E is preferred, and in September and October T is the better measure. Both measures, T and E, account for similar determinants of silage maize growth. The latter, however, is more complex because it contains information on sub-daily radiation additionally to daily temperature (Hargreaves and Samani, 1985). It can be assumed that this additional information are averaged out using monthly values and monthly temperature becomes a close estimate of monthly E. This is in alignment with results on different time resolutions, which indicate that measures of evapotranspirative demand are highly correlated





with temperature extremes (Roberts et al., 2013; Lobell et al., 2013). Therefore, it is sufficient to account for temperature when simultaneously controlling for water supply (P, SMI) because it is easier to measure temperature data and there is a smaller chance of attenuation bias.

The extent of the model improvement by adding soil moisture anomalies varies across the months. This can be evaluated by
comparing the gray and black markers in Fig. 2. Including soil moisture anomalies only improves model fit to a little extent in May and July. In all the other months, large improvement can be made when additionally controlling for soil moisture. In the second half of the season, i.e. August and September, the models using only SMI have a similar or even lower BIC compared to all meteorology-only models.

These results indicate that soil moisture builds memory over the season that adds relevant information not integrated in the
meteorological variables. There are several reasons for this postulation remark. First, the seasonality of soil moisture must be considered. The fraction of the saturated soil changes over time and thus the base value for the index. For Germany, this seasonality is depicted in Fig. 4 in Samaniego et al. (2013). In March, soil water content is the highest while soils are usually driest in August and September. This also implies, that an agricultural drought has a lower absolute soil moisture in August and September compared to the preceding months. Second, the anomalies in the later months integrate information about the
water balance in the preceding months because of the persistent character of soil moisture (evident from the autocorrelation of the soil moisture indexes). For instance, extreme dry conditions during flowering and grain filling are reflected in a dry soil moisture anomaly in the second half of the agricultural season of silage maize. The observation, that the SMI represents additional information to the meteorology is also pronounced by the fact that the pairwise correlations including SMI are lower compared to any other combination of the exogenous variables (Table 2). Further, dry anomalies in the late part of the season
may indicate a long lasting water shortage condition, as soil moisture drought lasts over several month or potentially even years (Sheffield and Wood, 2011; Samaniego et al., 2013; Zink et al., 2016).

In summary, soil moisture anomalies improve the model fit in all model configurations. This is the case even though soil moisture is strongly affected by the penalty for additional parameters within the BIC. Further, the evidence of nonlinear effects in the meteorological terms is confirmed. The results also indicate that there is substantial seasonal variability in the impact
of exogenous variables. This is investigated further quantitatively in the next sections for the meteorological variables and soil moisture.

## 4.2  Quantitative Assessment: Coefficient of determination for models using different explanatory variables

In this and the next section (4.3), we present the quantitative results for the "full" model with polynomials of degree three of the variables temperature (T) and precipitation (P) in the meteorological term and additionally the soil moisture anomalies
(SMI). Using the same model configuration for each month allows the comparison of partial effects and ensures that the source of variation is the same within the meteorological term (Auffhammer and Schlenker, 2014). In this section, the coefficient of determination is employed to evaluate the share of the sample variation only explained by the exogenous variables. Additionally, it is used to assess the in-sample goodness of fit of the models 03 (SMI & P), 05 (SMI & T), 08 (P & T), and 09 (SMI & P & T), each using polynomials of degree three.





**Table 3.** Comparison of the adjusted Coefficient of Determination $R^2$. The results from the demeaning framework serve as reference to the ones obtained by Least Square Dummy Variable Regression (LSDV). The latter explicitly accounts for the fixed effects. Additionally model configurations without either T, P, or SMI are shown.

|  | May | June | July | August | September | October | Average | June - August |
|---|---|---|---|---|---|---|---|---|
| (a) Adjusted $R^2$ demeaning | 0.11 | 0.16 | 0.31 | 0.17 | 0.13 | 0.12 | 0.16 | 0.21 |
| (b1) Adjusted $R^2$ LSDV | 0.56 | 0.59 | 0.66 | 0.59 | 0.57 | 0.56 | 0.59 | 0.61 |
| (b2) ((b1) − (a)) / (a) in % | 409.1 | 268.8 | 112.9 | 247.1 | 338.5 | 366.7 | 290.5 | 209.6 |
| (c1) Adjusted $R^2$ no T | 0.07 | 0.13 | 0.28 | 0.16 | 0.08 | 0.08 | 0.13 | 0.19 |
| (c2) ((c1) − (a)) / (a) in % | −36.4 | −18.8 | −9.7 | −5.9 | −38.5 | −33.3 | −23.7 | −11.4 |
| (d1) Adjusted $R^2$ no P | 0.08 | 0.11 | 0.22 | 0.14 | 0.12 | 0.12 | 0.13 | 0.16 |
| (d2) ((d1) − (a)) / (a) in % | −27.3 | −31.3 | −29.0 | −17.6 | −7.7 | 0.0 | −18.8 | −26.0 |
| (e1) Adjusted $R^2$ no SMI | 0.07 | 0.08 | 0.30 | 0.11 | 0.06 | 0.07 | 0.11 | 0.16 |
| (e2) ((e1) − (a)) / (a) in % | −36.4 | −50.0 | −3.2 | −35.3 | −53.8 | −41.7 | −36.7 | −29.5 |

The coefficients of determination for two model settings are evaluated to show the ability of the meteorological term and the soil moisture anomalies to improve the in-sample goodness of fit of the full model: first, the model that only accounts for the exogenous variation which is derived by the demeaning framework (row (a) in Table 3); second, the least squared dummy variable model that explicitly accounts for both the exogenous variation and the fixed effect (row (b1) in Table 3). The ratio of

5 the coefficient of determination derived by these two model setups is investigated (row (b2) in Table 3) to quantify the share of variance explained only by the meteorological term and soil moisture anomalies. Expectedly, the exogenous variation in weather and soil moisture improves the model fit in all months, but the level of improvement varies. The month which gains the least in explanatory power when explicitly accounting for the fixed effects is July (+ 112.9 %), indicating that a large share of the variation in yield is explained by the exogenous variables. The month with the largest share of the variation explained

by only fixed effects is May, where 409.1 % of the explanatory power is added when explicitly accounting for the mean effect of the administrative districts.

The adjusted $R^2$ presented in this study explicitly including fixed effects for each month of the period June (0.59), July (0.66), and August (0.59) is comparable to related approaches. Urban et al. (2015b), who employed a comparable period to estimate their results, reported $R^2$ of 0.65 and 0.67 for a model that successfully accounts for the interaction between heat and

15 moisture for a 61 - 90 day period following sowing for Iowa, Illinois, and Indiana. Their study additionally employed time fixed effects which usually lead to higher $R^2$. The seminal approach employing extreme degree days (EDD, Schlenker and Roberts, 2009) reported $R^2$ between 0.77 and 0.78. In their sample, a comparatively large share of the variation was explained by the fixed effects and trend, which exhibited an $R^2$ of 0.66. A study using updated data of Schlenker and Roberts (2009) and controlling for evaporative demand in July and August achieved adjusted $R^2$ between 0.66 and 0.72 (Roberts et al., 2013).



In the previous section, all the models have been evaluated with respect to the BIC criterion which penalizes over-fitting. The focus here is on reducing the sample bias of the model. The in-sample adjusted $R^2$ of the models is additionally compared when either one of the variables SMI, P, or T is not considered (rows (c1) - (e1) in Table 1). The according relative change in model fit when one variable is removed from the full model can be found in rows (c2) - (e2) of Table 3. In all months but

May and July, the strongest loss in in-sample goodness of fit is seen for removing soil moisture (for instance - 50.0 % in June and - 35.3 % in August). In July, which is the month with the highest overall in-sample-goodness of fit, the largest effects is accounted for by precipitation (- 29.0 %). The average relative model loss is largest for soil moisture for the entire season (-36.7 %) as well as the period June to August (-29.5 %). As observed in the section before, the effect of each particular variable is dependent on the month. For instance, the largest relative loss in adjusted $R^2$ for SMI is estimated in June (- 50.0 %) and

September (- 53.8 %). The largest effect of precipitation is observed in June (- 31.3 %) and July (- 29.0 %). Temperature is relevant the most in September (- 38.5 %) and May (- 36.4 %).

To summarize, the in-sample explanatory power of the full models are comparable to those reported in the previous literature. The largest average gain in goodness of fit is achieved by including SMI. In July, the month with the largest in-sample goodness of fit, most of the variation in yield is explained by precipitation. This section has only presented a quantitative analysis of

the predictive power in terms of adjusted $R^2$. A detailed assessment of the partial functional form of individual explanatory variables is presented in the next section to better understand their ceteris paribus impact on the crop yield.

### 4.3    Quantitative Assessment: Partial Effects of the Meteorological Variables

A better understanding of the relationship between individual explanatory variables allows to design effective adaptation measures. The partial functions of the meteorological covariates are presented in the next two sections and those of soil moisture

in section 4.3.3. Those functional forms, which are significant at least in the first or second order, are presented for individual months in Fig. 3. The range of the meteorological variables is depicted from - 2 to + 2 standard deviations (SD). It can be assumed that larger deviations from the mean are related to higher uncertainties in the estimated crop yield. A table with the estimated coefficients and standard errors of all models can be found in Table 4.

### 4.3.1    Partial Effects of Precipitation

The partial precipitation effects for the months May to August are shown in Panel a) of Fig. 3. Given constant soil moisture and temperature effects, negative precipitation anomalies are associated with reduced yield in these months. The largest effect is observed for June (- 5 % at - 1 SD) and July (- 6.5 % at - 1 SD). These are the overall most significant months, but with different patterns compared to the remaining two. In June and July, more than average precipitation is associated with comparatively higher yield (at 1 SD: + 2.2 % in June and + 2.1 % in July), whilst the opposite is the case for May and August.

The results indicate the importance of sufficient water supply provided to plants by precipitation, especially in June and July. In Germany, the begin of flowering is usually in July and extends into August (based on data provided by the German Weather Service - Deutscher Wetterdienst, 2017). Maize plants are susceptible to water stress during this growing phase (Barnabás et al., 2008; Fageria et al., 2006; Grant et al., 1989; Bolaños and Edmeades, 1996). Despite the necessity to control for intra-seasonal




**Table 4.** Results of Regression Models employing precipitation and temperature to account for meteorology (both with polynomials of degree 3, superscripts denote the degree of individual polynomials) and a stepwise function of SMI.

| | Dependent Variable: log(Silage Maize) | | | | | |
|---|---|---|---|---|---|---|
| | Model of the month | | | | | |
| | May | June | July | August | September | October |
| Precipitation[1] | 0.004 | 0.036*** | 0.039*** | −0.014 | −0.011 | −0.003 |
| | (0.011) | (0.014) | (0.013) | (0.011) | (0.013) | (0.010) |
| Precipitation[2] | −0.023* | −0.014* | −0.023*** | −0.019*** | −0.005 | 0.002 |
| | (0.014) | (0.007) | (0.004) | (0.006) | (0.005) | (0.008) |
| Precipitation[3] | 0.004 | 0.001 | 0.005*** | 0.004*** | 0.002 | −0.0001 |
| | (0.002) | (0.001) | (0.002) | (0.002) | (0.001) | (0.002) |
| Temperature[1] | 0.024 | −0.006 | −0.036* | −0.003 | 0.038 | −0.002 |
| | (0.021) | (0.015) | (0.021) | (0.014) | (0.024) | (0.018) |
| Temperature[2] | −0.005 | −0.006 | −0.007*** | −0.008** | −0.009* | −0.016** |
| | (0.007) | (0.006) | (0.002) | (0.003) | (0.005) | (0.008) |
| Temperature[3] | 0.0004 | −0.002 | 0.004* | −0.002 | −0.013* | 0.005 |
| | (0.003) | (0.003) | (0.003) | (0.002) | (0.006) | (0.003) |
| SMI: severe drought | 0.068*** | 0.024 | −0.044** | −0.110*** | −0.126*** | −0.149*** |
| | (0.012) | (0.020) | (0.019) | (0.035) | (0.028) | (0.037) |
| SMI: moderate drought | 0.044*** | 0.016 | −0.007 | −0.055*** | −0.041* | −0.024 |
| | (0.011) | (0.017) | (0.011) | (0.017) | (0.023) | (0.030) |
| SMI: abnormal dry | 0.011 | 0.023*** | −0.005 | −0.024** | −0.017 | −0.005 |
| | (0.011) | (0.007) | (0.007) | (0.011) | (0.015) | (0.017) |
| SMI: abnormal wet | −0.007 | −0.034*** | −0.011 | 0.026*** | 0.007 | −0.006 |
| | (0.014) | (0.011) | (0.007) | (0.008) | (0.011) | (0.019) |
| SMI: abundant wet | −0.014 | −0.052** | −0.004 | 0.027*** | 0.012 | −0.001 |
| | (0.020) | (0.025) | (0.009) | (0.008) | (0.017) | (0.015) |
| SMI: severe wet | −0.009 | −0.202*** | −0.041*** | 0.037*** | 0.030 | 0.025 |
| | (0.019) | (0.047) | (0.016) | (0.013) | (0.027) | (0.017) |
| Observations | 5,376 | 5,376 | 5,376 | 5,376 | 5,376 | 5,376 |
| $R^2$ | 0.113 | 0.173 | 0.326 | 0.179 | 0.136 | 0.129 |
| Adjusted $R^2$ | 0.105 | 0.162 | 0.305 | 0.168 | 0.127 | 0.121 |
| F Statistic | 53.151*** | 87.531*** | 203.025*** | 91.409*** | 65.891*** | 62.296*** |

*Note:* *p<0.1; **p<0.05; ***p<0.01

variability of precipitation effects, explicitly controlling for this sensitive phase is not very common in recent reduced form studies (Carleton and Hsiang, 2016). Notable exceptions are Lobell et al. (2011a), who used precipitation centered around flowering (anthesis) in statistical models based on historical data of trials in Africa, and Ortiz-Bobea and Just (2013), who





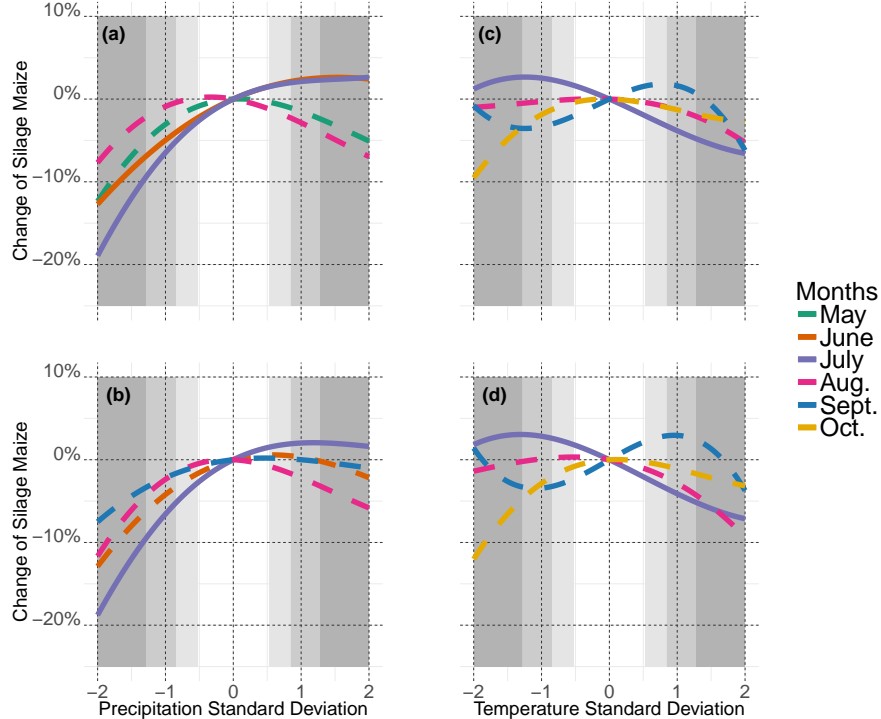

**Figure 3.** The partial dose-response functions of the meteorological variables are depicted for the range between - 2 and + 2 standard deviations (SD). The upper row represents those models considering SMI, whilst the lower row neglects SMI. A solid line is used for those variables which are significant in both the first and second degree polynomials. A dashed line is employed if only one of the first two polynomials is significant. The vertical axis represents the change in silage maize converted into % approximated by the formula $100(\exp(\sum_{j=1}^{3} \beta_j(\mathrm{x_{ikm}})^j) - 1)$, where $x_{ikm}$ is either precipitation or temperature. Under the assumption that the variables are normally distributed, the range depicted accounts for about 95 % of the observations. The dark gray areas denote the interval between the 0.023 % (- 2 SD) and the 10 % as well as the 90 % and 97.7 % (+ 2 SD) quantile. Similar, in medium gray the range between either the 10 % and the 20 % and the 80 % and 90 % quantiles is marked. The light gray quantifies the impact between the between either the 20 % and the 30 % and the 70 % and 80 % quantiles.

controlled for the vegetative, flowering, and grain-filling stages. Instead, many approaches employ total precipitation over the growing season (Annan and Schlenker, 2015; Burke and Emerick, 2016; Roberts et al., 2013; Schlenker and Roberts, 2006, 2009), monthly mean growing season precipitation (Urban et al., 2012) or the average of a subset of the season (Urban et al., 2015a). Studies for Germany commonly separate the season into the periods May to July and August to October (Gornott and Wechsung, 2015, 2016; Conradt et al., 2016), thus dividing exactly the time interval most susceptible to water stress and averaging over periods with diverse effects (e.g. May and June in Fig. 3a). This may hide water related effects. Other studies neglect precipitation entirely and only rely on temperature measures (Butler and Huybers, 2013, 2015; Schlenker et al., 2013).





According to their results, the explanatory power is not improved when adding precipitation. This is contradictory to our observations that precipitation is particularly relevant (see also Section 4.1 & 4.2).

The models employed here do not explicitly account for interactions between the meteorological and the soil moisture terms. Nevertheless, soil moisture is a function of the meteorological variables and all effects are correlated to each other (see Table 2).

The overall pattern in the effects of the meteorological variables only changes to a small extent when estimating the standard model configuration without the term for soil moisture anomalies (Fig. 3b). One of the most pronounced differences is that the positive effect of precipitation in June diminishes when not accounting for soil moisture. The coefficients in June are also less significant. The effects in September become significant in the second and third polynomial degree when not considering SMI (blue dashed line in Fig. 3b). On the contrary, May is less significant and thus not included in this panel. SMI improves

the model fit but only slightly affects the functional form of precipitation, which highlights that soil moisture adds relevant but different information as those entailed in precipitation. The next section presents an analogue analysis for temperature.

### 4.3.2   Partial Effects of Temperature

The significant partial temperature effects are depicted in Fig. 3c. A significant effect in all polynomials is only estimated for July, whilst in May and June, no significant coefficients can be found at all. In all months but September, higher than average

temperatures are associated with reduced crop yield. The extent of the effects, however, varies over time. In July, less than average temperature is associated with above-normal crop yield. The estimated function peaks at - 1.24 SD, which is 16.18 °C (mean in July is 18.34 °C). Additional 2.66 % crop yield can be expected at this temperature, all other variables hold constant. In August, elevated temperatures are associated with negative effects. September exhibits a large but not significant linear effect, whilst the second and third polynomials are significant. Because maize is maturing during this time, higher temperatures up

to a threshold are favorable as shown in Fig. 3c. Crop yield is reduced beyond this threshold, which might be related to heat waves. Cold temperatures have a negative effect in October, which is the strongest one observed. Harvesting commonly begins at the end of September within the period from 1999 to 2015 (Deutscher Wetterdienst, 2017). Thus, low temperatures may be related to early harvesting and result in lower yield.

When comparing the effects of precipitation and temperature in the months most relevant for meteorology, i.e. June and July,

those of precipitation clearly outweigh temperature. The largest effects can be found for negative anomalies of precipitation in July (compare Fig. 3a and Fig. 3c). The limited effect of temperature is in alignment with agricultural literature, which states that maize is tolerant to heat as long as enough water is provided (FAO Water, 2016). This is also the case in our study area given the fact that Germany lies in a rather temperate and marine climate zone. Additionally, sufficient provision of water is associated with prolonged grain filling and hence diminished heat sensitivity (Butler and Huybers, 2015). Recent literature

often neglected precipitation and emphasized mostly extreme temperature instead (Carleton and Hsiang, 2016; Lobell et al., 2008, 2011b; Schlenker et al., 2005; Schlenker and Lobell, 2010), which may have lead to biased assessments.

The general functional form of temperature are hardly affected by neglecting SMI (Fig. 3d). For example, crop yield changes from one - 3.82 % with SMI to - 4.11 % without for one SD of elevated temperature in July. These effects are smaller than





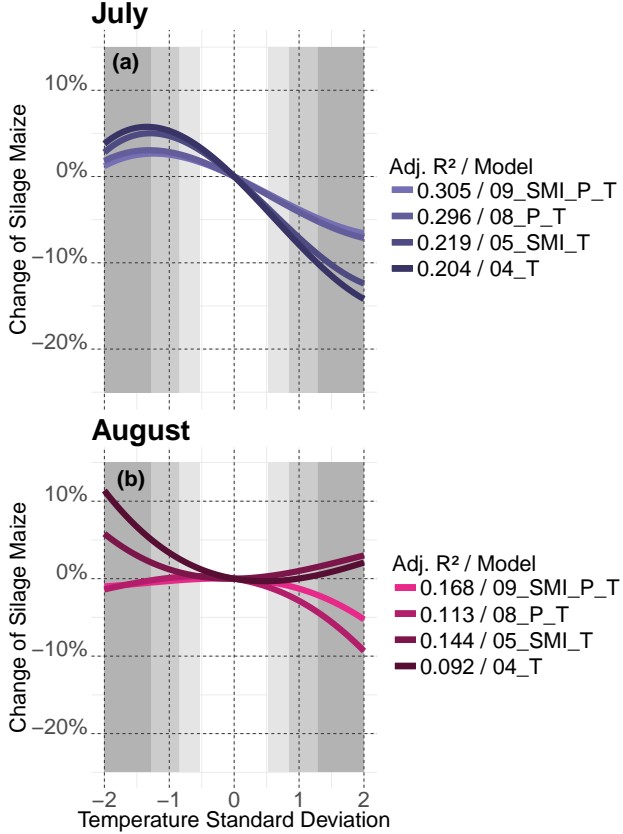

**Figure 4.** Sensitivity of the functional form of temperature partial effects for various controls for water supply.

those seen for precipitation, which highlights again that soil moisture provides an information that is independent to the one provided by T.

As mentioned before, a substantial amount of studies employed temperature as the major explanatory variable neglecting knowledge about plant physiology and plant growth (Wahid et al., 2007; FAO Water, 2016). The functional form of the partial

5   temperature effects derived from different model configurations for July and August is presented in Fig. 4 to evaluate the magnitude of bias between the full model (presented in Fig. 3) and a temperature-only model.

In both months, the in-sample explanatory power is reduced compared to the full model when only using temperature as explanatory variables. In July, the model fit is - 34.2 % lower when employing the temperature only model compared to the full model, while it is - 45.9 % in August (Fig. 4). In July, the in-sample goodness of fit is affected stronger by removing

10   precipitation (- 29.0 %) than by doing so for SMI (- 3.2 %), (Table 3). This is not surprising because the partial effect of precipitation in July is largest, whilst soil moisture anomalies only show negligible effect. On the contrary, considering SMI in August (- 35.3 %) exceeds the losses in Adjusted $R^2$ compared to a model without precipitation (- 17.6 %), (Table 3). In July, the functional form stays qualitatively the same across all model configurations (Fig. 4a). The magnitude of the effects





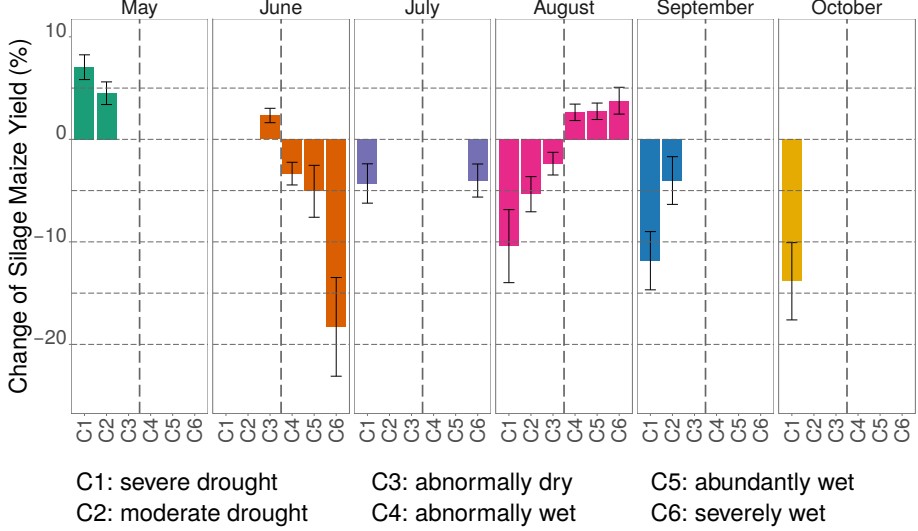

**Figure 5.** Percentage Change of Silage Maize Yield caused by significant Soil Moisture Anomalies for each month. The vertical axis represents the change in silage maize converted into % approximated by the formula $100(\exp(\sum_{j=1}^{6} \alpha_j \mathrm{I}(\mathrm{SMI}_{ikm} \in \mathrm{C_j}) - 1)$ , where $C_j$ are the soil moisture classes. The standard errors are indicated by the black error bars.

is, however, larger when precipitation is not considered. In August, the temperature effect is elevated by not considering SMI. Taking out precipitation reverses the effects found for the full models. This observation clearly demonstrates that adequate control of water supply is necessary to derive non-biased estimates of partial temperature effects. These results also indicate that the biases seen for different model configuration depend on the month considered. Overall, a model using only temperature

as explanatory variable has larger partial effects and potentially even different ones with regard to the direction compared to those of the full model. In the next section, the partial effects of the soil moisture index are investigated.

### 4.3.3    Partial Effects of the Soil Moisture Index (SMI)

Similar to the meteorological terms, the susceptibility to SMI changes over the months (Fig. 5). In particular, a change in the general patterns can be observed. In May and June, dry conditions are associated with positive yield (up to + 7 % in May,

and + 2.3 % in June), whilst wet conditions are harmful (up to - 18.3 %  under severely wet conditions in June). In July, both extremes have negative impacts of around - 4 %. In all of the following months, dry conditions are associated with reduced crop yield (up to - 10.4 % in August, - 11.8 % in September, and - 13.8 % in October), whilst only extreme wet conditions in August are positive for annual silage maize yield (up to + 3.77 %). These deviations are as high as the ones observed for the meteorological variables (Fig. 3).

For the interpretation of the results, the climatology of mean soil water content needs to be taken into account. The SMI of each month refers to different fractions of absolute water saturation in the soil. This seasonality is depicted in Fig. 4 in Samaniego et al. (2013) for different locations in Germany. In general, the optimal water content for plant development is





defined by 60 % to 80 % of the available field capacity, whilst less than 40 % field capacity, as for instance in the year 2003, is associated with depression in crop yield (Chmielewski, 2011). In May and June, dry anomalies represent soil moisture fractions above critical water content because the soil has been replenished with water in preceding winter and spring. For silage maize, however, rather dry conditions are preferable during this time because high soil moisture saturation can induce

luxury consumption and thus reduced root depths (FAO Water, 2016). This is particularly relevant for maize due to its capability to develop deep roots (FAO Water, 2016). This feature allows the plants to access deep soil water under dry conditions during the sensitive phase of flowering and grain filling. Empirical studies indicated that early wet conditions slow down the spreading of seeds and young plants can be damaged through indirect effects, such as fungus (Urban et al., 2015a). A detailed analysis indicates that the large effect of severely wet conditions in June can be partly associated to the 2013 flood in Germany (not

shown), which exhibited wet soils in large parts of the country. Starting in July, the level of soil water content decreases (see Fig. 4 in Samaniego et al., 2013). As a consequence, dry anomalies represent damaging conditions because plant available soil water starts to be too low to provide enough water during the most susceptible phase. These effects are increasing over the subsequent months because of the seasonality, the particular growing stage, and the persistence of soil moisture. Lower levels in absolute soil water also explain why wet anomalies have a positive impact in August, but not in July. July exhibits

the highest evapotranspiration among all months. This leads to a highly dynamic soil moisture in July which is characterized by a transition from a wet regime to a dry regime. Thus, small deviations from average soil moisture in this month have no significant effect on yield (Fig. 5). These are only observed for the very extreme conditions.

Additionally, the growing stage modifies the impact of soil moisture coefficients. In our sample, flowering commonly begins between mid- and end-July and milk ripening occurs in the second half of August (based on own calculation from data

provided by Deutscher Wetterdienst, 2017). Plants exhibit an increased susceptibility to insufficient water supply during these development stages. As shown in section 4.3, July has the highest partial effect with respect to meteorological variables. In August, soil moisture anomalies show a significantly higher impact on annual silage maize yield than in July. Due its seasonality, absolute soil moisture values are in general lower in August than in July. Further, soil moisture in August integrates temperature and precipitation effects of the preceding months. Thus, dry soil moisture anomalies show harmful effects, while

wet ones are beneficial. In September and October, soil moisture usually starts to refill (see Fig. 4 in Samaniego et al., 2013). Maize is in the less susceptible phase to dryness of ripening in September and harvesting usually starts in the second half of this month (Deutscher Wetterdienst, 2017). This implies, that severe drought anomalies in September and October might be associated with extended periods of water stress over the sensitive growing stages in the months before.

In this section, it was shown that the seasonality of soil moisture underlying the soil moisture index needs to be considered

to disentangled its temporal effects on silage maize yield. Thus, it is necessary to consider seasonality in soil moisture content and silage maize growth when assessing effects caused by soil moisture anomalies.



## 5 Conclusions

In this study, the intra-seasonal effects of soil moisture on silage maize yield in Germany are investigated. It is also evaluated how approaches considering soil moisture perform compared to meteorology-only ones. A demeaned reduced form panel approach is applied, which employs polynomials of degree three for variables of average temperature, potential evapotranspiration, precipitation, and a step wise function for soil moisture anomalies to capture nonlinearities. Potential evapotranspiration and average temperature are mutually exclusive. The model selection is based on the Bayesian Information Criterion (BIC) and the adjusted coefficient of determination ($R^2$).

This study provides a proof of concept, that a) soil moisture improves the capability of models to predict silage maize yield compared to meteorology-only approaches and that b) temporal patterns in the seasonal effects of the explanatory variables matter. It is shown that soil moisture anomalies improve the model fit in all model configurations according to both the BIC and $R^2$. SMI entails the highest explanatory power in all months but May (most explained by T) and July (most explained by P). This highlights that soil moisture adds different information than meteorological variables. All time invariant variables show seasonal patterns in accordance to each particular growing stage of silage maize. Furthermore, the dynamic patterns of the SMI effects originate from the seasonality in absolute soil moisture. Those results support the supposition that it is necessary to control for intra-seasonal variability in both the index for soil moisture and meteorology to derive valid impact assessments. Also, the comparison of various meteorological effects based on BIC showed that potential evapotranspiration adds no explanatory power compared to average temperature. Further, partial effects of precipitation outweigh those of temperature when controlling for intra-seasonal variability.

Our results have further implications for climate change impact assessment. First, it is shown that soil moisture can improve agricultural damage assessment and enrich the climate adaptation discourse in this realm, which is mostly based on temperature measures as major explanatory variable (Carleton and Hsiang, 2016). We recommend to control for at least those seasonal dependent pathways that affect plant growth presented in our study. Measures of soil moisture should be considered to derive evidence about climate impacts and adaptation possibilities. This particularly concerns climate econometrics, where frequently used reduced form approaches and dose-response functions should also control for soil moisture. For example, Butler and Huybers (2013) derived from a dose-response function only relying on temperature measures that the sensitivity to extreme degree days is lower in southern rather than northern U.S. counties. Based on these estimates they concluded that the south is better adapted to hot condition compared to the north. Transferring those adaptation potential to future impacts diminishes the estimated losses. However, various issues need to be considered when employing such an approach, such as the costs of adaptation and wrong institutional incentives (Schlenker et al., 2013; Annan and Schlenker, 2015). Also, Schlenker et al. (2013) argued that higher average humidity levels in the south diminish the correlation between heat and measures based on evapotranspirative demand. Accordingly, it is recommended to directly control for evapotranspirative demand by vapour pressure deficit (VPD). As shown in section 4.1, no superior effect of potential evapotranspiration over temperature was found when controlling for either precipitation or both precipitation and SMI. Potential evapotranspiration and VPD both account for the water demand of the atmosphere. Instead, the results of this study show that controlling for water supply by measures of





either soil moisture and precipitation avoids biased effects in a humid climate. This study further indicates, that it is necessary to account for the seasonal dynamics in both the meteorological and soil moisture effects that constitute the variation in crop yield to employ spatial adaptation as surrogate for future adaptation.

Second, the definition of an index as anomaly has general implications for climate econometrics. Such an index is less prone to systematic errors (Lobell2013, Gornott2015, Gornott2016), because any bias associated to the spatial processing and the meteorological or climatological modeling is minimized (Auffhammer et al., 2013; Conradt et al., 2016; Lobell, 2013). Also, the persistence in soil moisture and the resulting smoother distribution in comparison to the meteorological variables might deliver more reliable estimates than climate assessment based on meteorological variables because climate simulations only show robust trends at coarse temporal resolutions (Gornott and Wechsung, 2015). An index can also be interpreted as inter-annual variability beyond the demeaning framework. Any linear model employing a categorical variable for each spatial unit is equivalent to joint demeaning of both the dependent and the independent variables and thus the source of variation is the deviation from the mean. For instance, anomalies are used within the adaptation discourse to derive implications for short-term measures (Moore and Lobell, 2014). Again, in such a setting soil moisture can serve as more comprehensive measure than the commonly used temperature.

Finally, this study has also several implications for the design of adaptation measures on weather realizations to reduce current welfare losses of climate events (UNISDR, 2015; Kunreuther et al., 2009). First, indexes derived from soil moisture can be used in risk transfer mechanism. For instance, insurance schemes based on a particular weather indexes can be enhanced in both developed and developing countries (Agriculture Risk Management Team, 2011). Second, the detrimental effects of wet soil moisture anomalies might allow to extent the risk portfolio of multi-peril crop insurance and thus foster the advancement and implementation of those schemes in Germany (Keller, 2010). Third, the installation of agricultural infrastructure should be investigated because negative effects of soil moisture anomalies can be mitigated by irrigation and drainage. In 2010, only 2,34 % of the agricultural area used for silage maize is irrigated (own calculation from data provided by Statisitisches Bundesamt (2011)) and the latest numbers about drainage systems in Germany date back to 1993 (ICID, 2015).

Overall, an index of soil moisture considering intra-seasonal variability has relevant implications for current and future damage assessment and adaptation evaluation, which are supposed to gain importance in the course of climate change.

*Acknowledgements.* **We kindly acknowledge the German Meteorological Service (DWD), the Joint Research Center of the European Commission, the European Environmental Agency, the Federal Institute for Geosciences and Natural Resources (BGR), the Federal Agency for Cartography and Geodesy (BKG), the European Water Archiv, the Global Runoff Data Centre at the German Federal Institute of Hydrology (BfG), and the Federal Statistical Office of Germany for the provision of data. We especially thank Matthias Zink (UFZ) for processing and providing the data. We also regard thanks to the authors of the R-packages used in this study (Arnold (2016); Bivand et al. (2013, 2016); Bivand and Lewin-Koh (2017); Croissant and Millo (2008); Wickham (2007, 2011, 2016); Hlavac (2015); Hijmans (2016); Neuwirth (2014); Pierce (2015); Sarkar (2008); Sarkar and Andrews (2016)). We express our thanks to Prof. Reimund Schwarze for his comments and the promotion of the project at the Helmholtz Alliance Climate Initiative REKLIM. This work is also part of the Integrated Project Water Scarcity at the UFZ - Helmholtz Centre for Environmental Research, Leipzig,**





**Germany, which served as forum to present our work. Special thanks to Dr. Andreas Marx, head of the Climate Office for Central Germany, who supported us in the final steps of this study.**



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
