# Peer review of "The Effect of Soil Moisture Anomalies on Maize Yield in Germany"

_Natural Hazards and Earth System Sciences, 2017_

## Referee Comment (RC1) · Anonymous Referee #1 · 7 Jun 2017

The paper has in general excellent quality. Methods and analysis of results are well developed and sound. The only comment is that according to the underlying data bases (as well as indirect influence of spatial variability of soil conditions and soil+crop management) some more informaton should be provided regarding related uncertainties and limitations. For example, the potential influence of various maize cultivars used over Germany (on results) - through their different sensitivity to drought and heat stress - and related limitations as well as suggestions for future potential improvements of the demonstrated method in that context should be addressed, i.e. by application of high resolution Remote sensing data (Sentinel etc..).

---

## Referee Comment (RC2) · Anonymous Referee #2 · 6 Nov 2017

The manuscript "The effect of soil moisture anomalies on maize yield in Germany" by Peichl et al. describes the intra-seasonal impact of meteorological drivers and soil moisture on silage maize yields in Germany. Reduced form fixed effect models are employed to perform the analysis. The results are revealing the important meteorological factors driving inter-annual maize yield variability, and therefore provide important step towards development of seasonal forecasting framework (which can be used to advise farmers/policy makers). The results are interesting and scientifically sound. Nevertheless, I would suggest to revise the clarity of the methods used as well as several discussion aspects before the manuscript is accepted for publication.

Specific comments:

Page 1, line 21: instead of yield, I would mention maize yield variation.

[Figure]

Page 2, line 18: temperature above which threshold? Do you mean the Heat degree days (i.e. above ∼30 deg. C) or base temperature? Moreover, is temperature accumulation above that threshold considered or number of days with temperature above that threshold?

Page 2, line 20: I propose better reasoning before nonlinearity is mentioned; only stating the threshold is not sufficient. A reference to non-linear response of processes would be better.

Page 3, lines 1-5: Authors state that the temperature sensitivity is the highest during the flowering period. Nevertheless, high temperatures during the reproductive period affect maize as well, causing increased senescence rates, shortened duration of grain filling period and therefore reduced yields.

Page 5, line 15: rephrase (i.e. E is calculated based on empirical estimate, it is not measured)

Page 5, lines 28-30: Are crop yield data normalized as well? Moreover, you mention the normalization of meteorological data, followed by mentioning that their mean and standard deviation are 0 and 1, respectively. Does this mean that you have standardized rather than normalized the data? With this respect, the SMI ranges between 0 and 1, and is not standardized. How does this affect the regression coefficients? Better explanation would be welcomed.

Page 7, line 1: absence instead of absent

Page 7, Eq. 3: Index j in the first term indicates drought class, in the second term it indicates exponent and is therefore misleading. Moreover, three numberings are provided for the same equation.

Page 7, line 20: Why are coefficients constrained to be the same? Should they not reflect also the impact of non-climatic factors (such as agro-management, which can differ among different regions under consideration) on yield variability?

Page 7, lines 28-30: Better explanation is needed: - are fixed effect terms included in $C_i$? - how is $C_i$ calculated? - this substantially increases the number of variables for regression model trained on the relatively small sample sizes. It is worthwhile mentioning that standardization of predictors avoids problems with multi-colinearity arising from structure of regression models.

Page 8, line 1: what is natural experiment?

Page 9, line 1: what is demeaning? Short description might help.

Page 10, line 14: Is BIC related to explanatory power? Or is it referred to table 2?

Page 10, lines 15-20: What do non-linear terms represent? What could be the bio-physical reasoning behind?

Page 10, lines 20-25: In this analysis you have applied separately for each month the regression models an evaluated their capability to predict yields. You argue that soil moisture during the grain filling is a variable with "memory" from previous months and is therefore explaining more variability. Wouldn't that be the case also if you would take seasonal cumulates of precipitation and/or evapotranspiration, or climatic water balance (as their difference)? In that sense I agree that soil moisture is more relevant variable, however the comparison is not fair towards the monthly meteorological cumulates.

Page 11, lines 5-10: see previous comment – could seasonally cumulated precipitation and evaporation give similar results? Can you comment on this?

Page 11, lines 9-20: The study doesn't really show what is the role of soil information (i.e. soil water holding properties). Are there regions where SMI plays more/less (spatially variable) role with respect to meteorological counterpart? In the case where soils are able to retain water, the climatic water balance, integrated from the beginning of the season or for specific period during the season (i.e. flowering-maturity), could perform equally well. I think this might be important message of this study, which is not

adequately shown in results.

Page 12, line 8: 409 % with respect to what?

Page 13, line 15: Predictive power or explanatory power? This study does not assess the out-of-sample prediction performance; therefore, I would characterize adj. R2 as explanatory power.

Can these models be used for silage maize yield seasonal forecasting? Out-of-sample validation would be necessary to determine the predictive power, especially due to the fact that relatively short time series are used to construct the regression model. Can you comment on that?

Page 16, lines 5-11: What could be the biophysical meaning behind the second and third terms being significant?

---

## Author Comment (AC1) · 18 Dec 2017

We thank the reviewers for their detailed comments, which helped us to further improve our manuscript. We plane provide a revised pdf with highlighted changes in blue additional to the final manuscript.

A point-by-point response can be found below. Comments are in *italic* font; answers are in **bold** font.

**Reviewer #1**

*The paper has in general excellent quality. Methods and analysis of results are well developed and sound. The only comment is that according to the underlying data bases (as well as indirect influence of spatial variability of soil conditions and soil+crop management) some more information should be provided regarding related uncertainties and limitations. For example, the potential influence of various maize cultivars used over Germany (on results) -through their different sensitivity to drought and heat stress - and related limitations as well as suggestions for future potential improvements of the demonstrated method in that context should be addressed, i.e. by application of high resolution Remote sensing data (Sentinel etc..).*

**We would like to thank the reviewer for her/his comments on our manuscript that helped to further improve it. In detail we plan to address the comments as follows. Regarding the maize cultivars, the basic assumptions and limitations of the models will be elaborated in greater detail. This includes mentioning two major assumptions. First, that the farmers optimized their production process, given their experience about a particular site, which also covers the choice of the variety. Second, it is assumed that the response of plants to inter-annual stressors is the same across all locations. Future potential improvements include the expansion of the model to different varieties to alleviate the second assumption mentioned above. Another improvement that we point to is using remote sensing data. With regard to future improvements, we plan to include this in the discussion section, as this note relates mainly to the underlying data. Here we point out that improvements can be achieved through data with higher temporal resolution, which better take into account certain growth stages.**

---

## Author Comment (AC2) · 18 Dec 2017

We thank the reviewers for their detailed comments, which helped us to further improve our manuscript. We plane provide a revised pdf with highlighted changes in blue additional to the final manuscript.

A point-by-point response can be found below. Comments are in *italic* font; answers are in **bold** font.

**Reviewer #2**

*The manuscript "The effect of soil moisture anomalies on maize yield in Germany" by Peichl et al. describes the intra-seasonal impact of meteorological drivers and soil moisture on silage maize yields in Germany. Reduced form fixed effect models are employed to perform the analysis. The results are revealing the important meteorological factors driving inter-annual maize yield variability, and therefore provide important step towards development of seasonal forecasting framework (which can be used to advise farmers/policy makers). The results are interesting and scientifically sound. Nevertheless, I would suggest to revise the clarity of the methods used as well as several discussion aspects before the manuscript is accepted for publication.*
**We would like to thank the reviewer for her/his comments on our manuscript that helped to further improve it. We are going to substantially revise the methods section and take into account several discussion aspects as outlined below in more detail.**

*[1]Page 1, line 21: instead of yield, I would mention maize yield variation.*
**Corrected accordingly.**

*[2] Page 2, line 18: temperature above which threshold? Do you mean the Heat degree days (i.e. above 30 deg. C) or base temperature? Moreover, is temperature accumulation above that threshold considered or number of days with temperature above that threshold?*
**Thank you for this comment. In this paragraph we cite the results of other authors. Regarding the threshold: it is estimated to be 29° Celsius and explicitly mentioned in the text. To clarify the applied method, the original sentence of the seminal paper is included here: "The new weather data include the length of time each crop is exposed to each one-degree Celsius temperature interval in each day, summed across all days of the growing season, all estimated for the specific locations within each county where crops are grown" (Schlenker and Roberts, 2009). In other words, the authors considered the exposure of the plants to all one degree temperature intervals between 0 and 42 degrees at hourly time steps. They report substantial reductions in yield above 29 degrees. We plan to clarify this point in the corresponding paragraph in the introduction.**

*[3] Page 2, line 20: I propose better reasoning before nonlinearity is mentioned; only stating the threshold is not sufficient. A reference to non-linear response of processes would be better.*
**Thank you for this comment. We would like to highlight here that, as seen in Schlenker and Roberts (2009), these thresholds generally describe that the relationship between yield variability and temperature is non-linear, in the sense that it is initially increasing and then decreasing. This reflects the assumption that optimal conditions exist for plant growth in each growth stage (Thompson, 1969). In the seminal paper, to which we refer here mostly, no processes are described explicitly to underpin nonlinearity (Schlenker and Roberts, 2009). When it comes to "dynamic physiological process of plant growth, seed formation, and yield" (Schlenker and Roberts, 2009), the authors refer to agronomic studies, that "use a**

rich theoretical model to simulate yields given daily and subdaily weather inputs, nutrient applications, and initial soil conditions" (Schlenker and Roberts, 2009). "A strength of simulation models is that they fully incorporate plant-growth theory. These models also incorporate the whole distribution of weather outcomes over the growing season. This differs markedly from earlier regression-based approaches that typically use average weather outcomes or averages from particular months and thus give biased estimates of nonlinear temperature effects" (Schlenker and Roberts, 2009). The goal of this study is to test for and then introduce nonlinear effects of key determinants into regression models. These regression models are often used to estimate climate impact as they have the advantage of good predictive power (Timmins and Schlenker, 2009). In such a context, however, "[a]ccurate estimation of nonlinear effects is particularly important when considering large, nonmarginal changes in temperatures, now expected with climate change" (Schlenker and Roberts, 2009). It is planned to clarify the application context and why nonlinearity matter in such a context in the revised text.

*[4]Page 3, lines 1-5: Authors state that the temperature sensitivity is the highest during the flowering period. Nevertheless, high temperatures during the reproductive period affect maize as well, causing increased senescence rates, shortened duration of grain filling period and therefore reduced yields.*
We thank the reviewer for this comment. In this section we try to show that the sensitivity to stressors such as temperature above 29 degrees is not the same for all growth stages, but strongly depends on the stage of growth. We use the flowering period as an example of such an increased susceptibility, amongst others. We plan to include the reproductive period as an example in the revised version, since it is also mentioned in the literature cited (see for instance Wahid et al. (2007)).

*[5] Page 5, line 15: rephrase (i.e. E is calculated based on empirical estimate, it is not measured.*
We agree with the reviewer on this comment. We plan to accordingly rephrase it in the revised section: "E is calculated by the equation of Hargreaves and Samani (1985) based upon extraterrestrial radiation and temperature and is estimated in millimeter per day."

*[6] Page 5, lines 28-30: Are crop yield data normalized as well? Moreover, you mention the normalization of meteorological data, followed by mentioning that their mean and standard deviation are 0 and 1, respectively. Does this mean that you have standardized rather than normalized the data? With this respect, the SMI ranges between 0 and 1, and is not standardized. How does this affect the regression coefficients? Better explanation would be welcomed.*
Thank you for that comment. Harvest yields are not normalized. However, it is logarithmized and adjusted linearly to the time trend (see section 2.1.). In addition, the fixed effects on the right-hand side of the equation 2 also take into account the mean yield for each administrative district. This means that the data for each district is mean-centered. However, the spread of returns remains unchanged. We agree on the terminology and now use standardization instead of normalization in the revised version. Regarding the standardization of SMI. SMI is already defined as a measure of an anomaly. An alternative would be to standardize soil moisture. However, it is unclear how drought can be defined in such a case. Here, we use SMI because it is a standard approach for quantifying soil moisture drought. See for example the German Drought Monitor and the US Drought Monitor. In addition, we use index functions and standardization would take no effects because the intervals would be adjusted accordingly. In other words the index functions would give the same results irrespective of the usage of SMI or Soil Moisture. The index functions are used to map soil moisture to six classes of dryness and wetness. This mapping

**uses location depended percentiles to guarantee a spatial consistency.**

*[7]Page 7, line 1: absence instead of absent.*
**Corrected.**

*[8] Page 7, Eq. 3: Index j in the first term indicates drought class, in the second term it indicates exponent and is therefore misleading. Moreover, three numberings are provided for the same equation.*
**Thanks for the remark. This is now corrected. We will use the index n for classes. In addition, the numbering is corrected and we will use only one for the same equation.**

*[9] Page 7, line 20: Why are coefficients constrained to be the same? Should they not reflect also the impact of non-climatic factors (such as agro-management, which can differ among different regions under consideration) on yield variability?*
**Thank you for your comment on our modeling approach. In general, we rely on methods that are widely used for studies in the United States and other parts of the world and are based on econometric theory, especially in the evolving field of climate econometrics. One of the basic assumptions is that we do not explicitly control endogenous processes such as management. Instead, only exogenous variation within the sample is used (Timmins and Schlenker, 2009). Instead, we control time-variant differences between the districts through constants. These are the mean values of the explained and explanatory variables specific to each district and add up to the fixed effects ($c_{im}$). Fixed effects are a common term in econometrics. Since fixed effects represent the means of a predictor such as meteorology and soil moisture for a certain period, it may be interpreted as the long term conditions for farming (Schlenker and Roberts, 2006). Those are usually known by the farmer. For instance, it might be assumed that the farmer does not only know the average weather conditions but also the water holding capacity of the soil and adapts the management accordingly. The source of identification of the coefficients is thus year-to-year variation of weather and soil moisture. This interpretation is enhanced by the use of anomaly categories for soil moisture. This is going to be clarified in more detail in section 3.**

*[10] Page 7, lines 28-30: Better explanation is needed: - are fixed effect terms included in Ci? - how is Ci calculated? - this substantially increases the number of variables for regression model trained on the relatively small sample sizes. It is worthwhile mentioning that standardization of predictors avoids problems with multi-collinearity arising from structure of regression models.*
**Thanks a lot for this comment which helps to clarify important aspects of the model. The fixed effects ($c_{im}$) are constants that are generated for each district and are composed of the administrative district specific means of the left and right variables of the model. It is possible to take these explicitly into account by using dummy variables for each district, here referred to as the least square dummy variable approach (lsdv). Alternatively, each variable can be demeaned for each district, here referred to as the demeaning framework. The demeaning framework therefore has fewer variables to be estimated. However, both approaches provide the same estimates for the coefficients. Also, the fixed are function of the means and the coefficients and thus add no degrees of freedom to the model. We plan to clarify the meaning of fixed effects in the revised text. For instance we plan to include: "Time-invariant differences between administrative districts are taken into account by the fixed effects. These consist of the districts specific mean values of the individual variables on the right and left side of the equation".**
**Some clarifications will be implemented also in the section, in which we compare least square dummy variable approaches and the demeaning framework. With regard to the standardization recommendation, we have not been able to find any evidence of it. We only found literature suggesting that centered means play a role in dealing with problems of multicollinearity caused by polynomial concepts. How-**

ever, we found no differences in the standard errors between those estimated by the demeaning framework, where the administrative specific mean for each spatial unit is subtracted, and the least squares dummy variable approach. Thus, centering by demeaning has no influence on the standard errors.

*[11]Page 8, line 1: what is natural experiment?*
In social science, where experiments are very difficult to implement, it describes a study design which allows to reveal causal effects. A typical example in econometrics is, that a "natural experiment occurs when some exogenous event - often a change in government policy - changes the environment in which individuals, families, firms, or cities operate. A natural experiment always has a control group, which is not affected by the policy change, and a treatment group, which is thought to be affected by the policy change. Unlike a true experiment, in which treatment and control groups are randomly and explicitly chosen, the control and treatment groups in natural experiments arise from the particular policy change" (Wooldridge, 2012). In our case, in which we rely only on exogenous variations of period-to-period effects, the farmer cannot choose which group they belong to, as the weather conditions are stochastic. This is will be clarified in more detail in section 3.

*[12] Page 9, line 1: what is demeaning? Short description might help.*
Demeaning is subtracting the mean, here the administrative specific mean of all variables. Description will be included in the revised text.

*[13] Page 10, line 14: Is BIC related to explanatory power? Or is it referred to table 2?*
The Bayesian information criterion is chosen by us as a measure to achieve a good balance between the explanatory power (i.e. goodness of fit in the sample) and model complexity (i.e. number of parameters). It is not referring to table 2. This table show the comparison of the Pearson Correlation Coefficients of the exogenous variables, in particular to show that E and T are highly correlated (e.g. as a measure of multicollinearity).

*[14] Page 10, lines 20-25: In this analysis you have applied separately for each month the regression models and evaluated their capability to predict yields. You argue that soil moisture during the grain filling is a variable with memory from previous months and is therefore explaining more variability. Wouldnt that be the case also if you would take seasonal cumulates of precipitation and/or evapotranspiration, or climatic water balance (as their difference)? In that sense I agree that soil moisture is more relevant variable, however the comparison is not fair towards the monthly meteorological cumulates.*
We performed additional analysis to investigate the relationship between seasonal cumulates of precipitation and soil moisture (see figure 1 attached). As expected longer cumulates of precipitation show a higher correlation with soil moisture as compared to one month precipitation. Soil moisture provides different information as compared to seasonal cumulates of precipitation, which becomes apparent from the following two observations. First, there is a strong spatial variability of the Pearson Correlation Coefficient with relatively lower values in Eastern Germany and higher values in the Southern Germany. Second, the mean coefficient of determination is at most 50 %. This implies that the substantial part of the soil moisture variability cannot be explained by precipitation, even at season accumulates. This stems from the fact that soil moisture persistence is not only determined by precipitation but also by evapotranspiration and soil hydraulic properties. Therefore, soil moisture has a qualitatively different memory as precipitation and we think it is fair to state that the model profits from the persistence of soil moisture. However, we will acknowledge in the revised manuscript that seasonal cumulates of precipitation do exhibit longer memory than one month precipitation.

*[15] Page 11, lines 5-10: see previous comment could seasonally cumulated precipitation and evaporation give similar results? Can you comment on this?*
**Please see answer to previous comment.**

*[16] Page 11, lines 9-20: The study doesn't really show what is the role of soil information (i.e. soil water holding properties). Are there regions where SMI plays more/less (spatially variable) role with respect to meteorological counterpart? In the case where soils are able to retain water, the climatic water balance, integrated from the beginning of the season or for specific period during the season (i.e. flowering-maturity), could perform equally well. I think this might be important message of this study, which is not adequately shown in results.*

**Thanks for this valuable comment. As can be seen in figure 1 (attached), spatially heterogeneity exist in the correlation between SMI and different measures of cumulated weather measures such as precipitation. To incorporate the same information as those used by soil moisture it would be necessary to take into account different accumulates of precipitation for different location. For example Southern Germany exhibits higher water retaining capacities and also higher correlation with three month precipitation as compared to Eastern Germany. However, a substantial share of the variability of soil moisture is not explained by precipitation (see also comment above). One advantage of using soil moisture in such a study is that we can restrict the coefficients to be the same at all locations, whilst assuming that the water retaining is not the same everywhere. We will add this point to the discussion of the results.**

*[17] Page 12, line 8: 409 % with respect to what?*
**In this context, 409.1 % refer to the relative difference of the adjusted R-square derived least squared dummy variable regression with respect to the one derived by the demeaning framework. In other words, it is the explanatory power added if the average yield of each county is explicitly taken into account in comparison to the models that only use soil moisture and weather variation as explanatory variables. We will further clarify this in the revised text.**

*[18] Page 13, line 15: Predictive power or explanatory power? This study does not assess the out-of-sample prediction performance; therefore, I would characterize adj. R2 as explanatory power. Can these models be used for silage maize yield seasonal forecasting? Out-of-sample validation would be necessary to determine the predictive power, especially due to the fact that relatively short time series are used to construct the regression model. Can you comment on that?*
**We agree with the reviewer that we only assessed the explanatory power of the model in this study and will adjust the test accordingly. It is planned to use the models of these studies for seasonal forecasting and climate projections in follow-up studies. As far as the first part of the commentary is concerned, we agree will use explanatory power instead. Part two: It is planned to use a combined version of this model for seasonal and climate forecasting in another study. In this study here, we used BIC to benchmark the model configurations, because it penalizes model over-fitting which hampers the out-of-samples predictability. However, follow up studies should perform an out-of-sample validation to determine the predictive power.**

*[19] Page 16, lines 5-11: What could be the biophysical meaning behind the second and third terms being significant?*
**We want to highlight here that it is very challenging to relate biophysical process to the coefficients of the statistical model used in this study. An approach using a processed-based crop model would be required to uncover these relationships. The main objective of considering non-linearities is to represent large and non marginal**

changes in the weather system (due to climate change) that are outside of the optimal plant growth conditions (we plan to acknowledge this in Section 1 & 4.1). Therefore, non-linear dose-response functions are implemented. Furthermore, in this setting the terms of the polynomials are not orthogonal to each other. Thus, the non-significance of a term cannot be equated with a test for non-linearity. On the basis of the study design, the common significance of all polynomials rather implies a causal effect.

**References**

Schlenker, W. and Roberts, M. J.: Nonlinear Effects of Weather on Corn Yields, Review of Agricultural Economics, 28, 391–398, doi:10.1111/j.1467-9353.2006.00304.x, 2006.

Schlenker, W. and Roberts, M. J.: Nonlinear temperature effects indicate severe damages to U.S. crop yields under climate change, Proceedings of the National Academy of Sciences, 106, 15 594–15 598, doi:10.1073/pnas.0906865106, 2009.

Thompson, L. M.: Weather and Technology in the Production of Corn in the U. S. Corn Belt, Agronomy Journal, doi:10.2134/agronj1969.00021962006100030037x, 1969.

Timmins, C. and Schlenker, W.: Reduced-Form Versus Structural Modeling in Environmental and Resource Economics, Annual Review of Resource Economics, 1, 351–380, doi: 10.1146/annurev.resource.050708.144119, 2009.

Wahid, A., Gelani, S., Ashraf, M., and Foolad, M. R.: Heat tolerance in plants: An overview, Environmental and Experimental Botany, 61, 199–223, doi:10.1016/j.envexpbot.2007.05.011, 2007.

Wooldridge, J.: Introductory econometrics: A modern approach, South-Western Cengage Learning, fourth edn., 2012.

**May**

[Figure]

Figure 1: Figure of the Pearson Correlation Coefficients of soil moisture of May and accumulated precipitation for one month (left panel: May), three months (middle panel: March to May), and six months (right pane:l: December to May)

---

## Author Response (AR1)

We thank the reviewers and the editor for their comments, which helped us to further improve our manuscript. We provide a marked-up manuscript with highlighted changes in blue additional to the final manuscript.

A point-by-point response to the reviewer can be found below. Comments are in *italic* font; answers are in **bold** font. The pages and lines of the changes are included and refer to the pdf file with highlighted changes.

The most relevant changes are as follows:

- We expanded the method section and clarified the use of fixed effect terms (page 8, lines 1 to 15 in the manuscript with highlighted changes).

- We included a paragraph discussing the use of seasonal cumulates of precipitation (page 11, line 32 to page 12, line 4 in the manuscript with highlighted changes).

**Reviewer #1**

*The paper has in general excellent quality. Methods and analysis of results are well developed and sound. The only comment is that according to the underlying data bases (as well as indirect influence of spatial variability of soil conditions and soil+crop management) some more information should be provided regarding related uncertainties and limitations. For example, the potential influence of various maize cultivars used over Germany (on results) -through their different sensitivity to drought and heat stress - and related limitations as well as suggestions for future potential improvements of the demonstrated method in that context should be addressed, i.e. by application of high resolution Remote sensing data (Sentinel etc..).*
**We would like to thank the reviewer for her/his comments on our manuscript that helped to further improve it. In detail we addressed the comments as follows. Regarding the maize cultivars, the basic assumptions and limitations of the models is elaborated in greater detail. This includes mentioning two major assumptions. First, that the farmers optimized their production process, given their experience about a particular site, which also covers the choice of the variety. Second, it is assumed that the response of plants to inter-annual stressors is the same across all locations. Future potential improvements include the expansion of the model to different varieties to alleviate the second assumption mentioned above (changes are implemented on page 8, lines 1 - 15 in the marked-up manuscript). Another improvement that we refer to is using remote sensing data. With regard to future improvements, we include this in the discussion section, as this note relates mainly to the underlying data. Here we point out that improvements can be achieved through data with higher temporal resolution, which better take into account certain growth stages (changes are implemented page 21, lines 7 - 9 in the marked-up manuscript).**

**Reviewer #2**

*The manuscript "The effect of soil moisture anomalies on maize yield in Germany" by Peichl et al. describes the intra-seasonal impact of meteorological drivers and soil moisture on silage maize yields in Germany. Reduced form fixed effect models are employed to perform the analysis.*

*The results are revealing the important meteorological factors driving inter-annual maize yield variability, and therefore provide important step towards development of seasonal forecasting framework (which can be used to advise farmers/policy makers). The results are interesting and scientifically sound. Nevertheless, I would suggest to revise the clarity of the methods used as well as several discussion aspects before the manuscript is accepted for publication.*

**We would like to thank the reviewer for her/his comments on our manuscript that helped to further improve it. We revised the methods section and take into account several discussion aspects as outlined below in more detail.**

*[1]Page 1, line 21: instead of yield, I would mention maize yield variation.*
**Corrected accordingly on page 1, line 21 in the marked-up manuscript.**

*[2] Page 2, line 18: temperature above which threshold? Do you mean the Heat degree days (i.e. above 30 deg. C) or base temperature? Moreover, is temperature accumulation above that threshold considered or number of days with temperature above that threshold?*

**Thank you for this comment. In this paragraph we cite the results of Schlenker and Roberts (2009). Regarding the threshold: it is estimated to be 29° Celsius and is explicitly mentioned in the text. To clarify the applied method, the original sentence of the seminal paper is included here: "The new weather data include the length of time each crop is exposed to each one-degree Celsius temperature interval in each day, summed across all days of the growing season, all estimated for the specific locations within each county where crops are grown" (Schlenker and Roberts, 2009). In other words, the authors considered the exposure of the plants to all one degree temperature intervals between 0 and 42 degrees at hourly time steps. They report substantial reductions in yield above 29 degrees. We clarified this point in the corresponding paragraph in the introduction ( page 2, lines 18 - 24 in the marked-up manuscript) .**

*[3] Page 2, line 20: I propose better reasoning before nonlinearity is mentioned; only stating the threshold is not sufficient. A reference to non-linear response of processes would be better.*

**Thank you for this comment. We would like to highlight here that, as seen in Schlenker and Roberts (2009), these thresholds generally describe that the relationship between yield variability and temperature is non-linear, in the sense that it is initially increasing and then decreasing. This reflects the assumption that optimal conditions exist for plant growth in each growth stage (Thompson, 1969). In the seminal paper, to which we refer here mostly, no processes are described explicitly to underpin nonlinearity (Schlenker and Roberts, 2009). When it comes to "dynamic physiological process of plant growth, seed formation, and yield" (Schlenker and Roberts, 2009), the authors refer to agronomic studies, that "use a rich theoretical model to simulate yields given daily and subdaily weather inputs, nutrient applications, and initial soil conditions" (Schlenker and Roberts, 2009). "A strength of simulation models is that they fully incorporate plant-growth theory. These models also incorporate the whole distribution of weather outcomes over the growing season. This differs markedly from earlier regression-based approaches that typically use average weather outcomes or averages from particular months and thus give biased estimates of nonlinear temperature effects" (Schlenker and Roberts, 2009). The goal of this study is to test for and then introduce nonlinear effects of key determinants into regression models. These regression models are often used to estimate climate impact as they have the advantage of good predictive power (Timmins and Schlenker, 2009). In such a context, however, "[a]ccurate estimation of nonlinear**

effects is particularly important when considering large, nonmarginal changes in temperatures, now expected with climate change" (Schlenker and Roberts, 2009). We clarified the application context and why nonlinearity matter in such a context in the revised text ( page **2**, lines **18 - 24** & page **10**, lines **9 - 14** in the marked-up manuscript).

*[4]Page 3, lines 1-5: Authors state that the temperature sensitivity is the highest during the flowering period. Nevertheless, high temperatures during the reproductive period affect maize as well, causing increased senescence rates, shortened duration of grain filling period and therefore reduced yields.*

We thank the reviewer for this comment. In this section we try to show that the sensitivity to stressors such as temperature above **29** degrees is not the same for all growing periods, but strongly depends on the stage of growth. We use the flowering period as an example of such an increased susceptibility, amongst others. We included the reproductive period as an example in the revised version, since it is also mentioned in the literature cited (see for instance Wahid et al. (**2007**)) (changes are implemented on page **3**, lines **4 - 5** in the marked-up manuscript).

*[5] Page 5, line 15: rephrase (i.e. E is calculated based on empirical estimate, it is not measured.*

We agree with the reviewer on this comment. We rephrased it in the revised section: "E is calculated by the equation of Hargreaves and Samani (**1985**) based upon extraterrestrial radiation and temperature and is estimated in millimeter per day" (page **5**, lines **19 - 20** in the marked-up manuscript).

*[6] Page 5, lines 28-30: Are crop yield data normalized as well? Moreover, you mention the normalization of meteorological data, followed by mentioning that their mean and standard deviation are 0 and 1, respectively. Does this mean that you have standardized rather than normalized the data? With this respect, the SMI ranges between 0 and 1, and is not standardized. How does this affect the regression coefficients? Better explanation would be welcomed.*

Thank you for that comment. Harvest yields are not normalized. However, it is logarithmized and adjusted linearly to the time trend (see section 2.1.). In addition, the fixed effects on the right-hand side of the equation **2** also take into account the mean yield for each administrative district. This means that the data for each district is mean-centered. However, the spread of returns remains unchanged. We agree on the terminology and now use standardization instead of normalization in the revised version (page **5**, line **30** in the final version; page **6**, line **1** in the marked-up manuscript). Regarding the standardization of SMI. SMI is already defined as a measure of an anomaly. An alternative would be to standardize soil moisture. However, it is unclear how drought can be defined in such a case. Here, we use SMI because it is a standard approach for quantifying soil moisture drought. See for example the German Drought Monitor and the US Drought Monitor. In addition, we use index functions and standardization would take no effects because the intervals would be adjusted accordingly. In other words the index functions would give the same results irrespective of the usage of SMI or Soil Moisture. The index functions are used to map soil moisture to six classes of dryness and wetness. This mapping uses location depended percentiles to guarantee a spatial consistency (lines **11 - 12** in the marked-up manuscript).

*[7]Page 7, line 1: absence instead of absent.*

Corrected on page **7**, line **4** in the marked-up manuscript.

*[8] Page 7, Eq. 3: Index j in the first term indicates drought class, in the second term it indicates exponent and is therefore misleading. Moreover, three numberings are provided for the same equation.*

Thanks for the remark. This is now corrected on page 7 (equation 2) and page 7, line 21 in the marked-up manuscript. We use the index n for classes. In addition, the numbering is corrected and we use only one for the same equation.

*[9] Page 7, line 20: Why are coefficients constrained to be the same? Should they not reflect also the impact of non-climatic factors (such as agro-management, which can differ among different regions under consideration) on yield variability?*

Thank you for your comment on our modeling approach. In general, we rely on methods that are widely used for studies in the United States and other parts of the world and are based on econometric theory, especially in the evolving field of climate econometrics. One of the basic assumptions is that we do not explicitly control for endogenous processes such as management. Instead, only exogenous variation within the sample is used (Timmins and Schlenker, 2009). Instead, we control for time-variant differences between the districts through constants. These are the mean values of the explained and explanatory variables specific to each district and add up to the fixed effects ($c_{im}$). Fixed effects are a common term in econometrics. Since fixed effects represent the means of a predictor such as meteorology and soil moisture for a certain period, it may be interpreted as the long-term conditions for farming (Schlenker and Roberts, 2006). Those are usually known by the farmer. For instance, it is assumed that the farmer does not only know the average weather conditions but also the water holding capacity of the soil and adapts the management accordingly. The source of identification of the coefficients is thus year-to-year variation of weather and soil moisture. This interpretation is enhanced by the use of anomaly categories for soil moisture. This is clarified in more detail in section 3 (page 7, line 24 - 25, and page 8, lines 1 - 15, in the marked-up manuscript).

*[10] Page 7, lines 28-30: Better explanation is needed: - are fixed effect terms included in Ci? - how is Ci calculated? - this substantially increases the number of variables for regression model trained on the relatively small sample sizes. It is worthwhile mentioning that standardization of predictors avoids problems with multi-collinearity arising from structure of regression models.*

Thanks you for this valuable comment which helps to clarify important aspects of the model. The fixed effects ($c_{im}$) are constants that are generated for each district and are composed of the administrative district specific means of the left and right variables of the model. It is possible to take these explicitly into account by using dummy variables for each district, here referred to as the least square dummy variable approach (lsdv). Alternatively, each variable can be demeaned for each district, here referred to as the demeaning framework. The demeaning framework therefore has fewer variables to be estimated. However, both approaches provide the same estimates for the coefficients. Also, the fixed effects are functions of the means and the coefficients and thus add no degrees of freedom to the dimensionality of the parameter space. We plan to clarify the meaning of fixed effects in the revised text. For instance we plan to include: "Time-invariant differences between administrative districts are taken into account by the fixed effects. These consist of the districts specific mean values of the individual variables on the right and left side of the equation" (page 7, lines 24 -25 in the marked-up manuscript. We also deleted a half sentence on page 9, line 3 in the marked-up manuscript).

Some clarifications are implemented also in the sections, in which we compare least

square dummy variable approaches and the demeaning framework (page 9, lines 14 - 15 & page 13, lines 1 - 16 in the marked-up manuscript).

With regard to the standardization recommendation, we have not been able to find any evidence of it. We only found literature suggesting that centered means play a role in dealing with problems of multicollinearity caused by polynomial concepts. However, we found no differences in the standard errors between those estimated by the demeaning framework, where the administrative specific mean for each spatial unit is subtracted, and the least squares dummy variable approach. Thus, centering by demeaning has no influence on the standard errors.

*[11]Page 8, line 1: what is natural experiment?*
In social science, where experiments are very difficult to implement, it describes a study design which allows to reveal causal effects. A typical example in econometrics is, that a "natural experiment occurs when some exogenous event - often a change in government policy - changes the environment in which individuals, families, firms, or cities operate. A natural experiment always has a control group, which is not affected by the policy change, and a treatment group, which is thought to be affected by the policy change. Unlike a true experiment, in which treatment and control groups are randomly and explicitly chosen, the control and treatment groups in natural experiments arise from the particular policy change" (Wooldridge, 2012). In our case, in which we rely only on exogenous variations of period-to-period effects, the farmer cannot choose which group they belong to, as the weather conditions are stochastic. This is clarified in more detail in section 3 (page 8, lines 10 - 13 in the marked-up manuscript).

*[12] Page 9, line 1: what is demeaning? Short description might help.*
Demeaning is subtracting the mean, here the administrative specific mean of all variables. Description is included in the revised text (page 9, lines 14 - 15 in the marked-up manuscript).

*[13] Page 10, line 14: Is BIC related to explanatory power? Or is it referred to table 2?*
The Bayesian information criterion is chosen by us as a measure to achieve a good balance between the explanatory power (i.e. goodness of fit in the sample) and model complexity (i.e. number of parameters). It is not referring to table 2. This table show the comparison of the Pearson Correlation Coefficients of the exogenous variables, in particular to show that E and T are highly correlated (e.g. as a measure of multicollinearity).

*[14] Page 10, lines 20-25: In this analysis you have applied separately for each month the regression models and evaluated their capability to predict yields. You argue that soil moisture during the grain filling is a variable with memory from previous months and is therefore explaining more variability. Wouldnt that be the case also if you would take seasonal cumulates of precipitation and/or evapotranspiration, or climatic water balance (as their difference)? In that sense I agree that soil moisture is more relevant variable, however the comparison is not fair towards the monthly meteorological cumulates.*
We performed additional analysis to investigate the relationship between seasonal cumulates of precipitation and soil moisture (figure 1 in this document). As expected longer cumulates of precipitation show a higher correlation with soil moisture as compared to one month precipitation. Soil moisture provides different information as compared to seasonal cumulates of precipitation, which becomes apparent from the following two observations. First, there is strong spatial variability of the Pearson

**May**

[Figure]

Figure 1: Figure of the Pearson Correlation Coefficients of soil moisture of May and accumulated precipitation for one month (left panel: May), three months (middle panel: March to May), and six months (right pane:l: December to May)

**Correlation Coefficient with relatively lower values in Eastern Germany and higher values in the Southern Germany. Second, the mean coefficient of determination is at most 50 %. This implies that a substantial part of the soil moisture variability cannot be explained by precipitation, even at the seasonal time scale. This stems from the fact that soil moisture persistence is not only determined by precipitation but also by evapotranspiration and soil hydraulic properties. Therefore, soil moisture has a qualitatively different memory as precipitation and we think it is fair to state that the model profits from the persistence of soil moisture. However, we will acknowledge in the revised manuscript that seasonal cumulates of precipitation do exhibit longer memory than one month precipitation (implemented on page 11, line 32 - page 12, line 4 in the marked-up manuscript).**

*[15] Page 11, lines 5-10: see previous comment could seasonally cumulated precipitation and evaporation give similar results? Can you comment on this?*
**Please see answer to previous comment.**

*[16] Page 11, lines 9-20: The study doesnt really show what is the role of soil information (i.e. soil water holding properties). Are there regions where SMI plays more/less (spatially variable) role with respect to meteorological counterpart? In the case where soils are able to retain water, the climatic water balance, integrated from the beginning of the season or for specific period during the season (i.e. flowering-maturity), could perform equally well. I think this might be important message of this study, which is not adequately shown in results.*
**Thanks for that valuable comment. As can be seen in figure 1 in this document, spatially heterogeneity exist in the correlation between SMI and different measures of cumulated weather measures such as precipitation. To incorporate the same information as those used by soil moisture it would be necessary to take into account different accumulates of precipitation for different locations. For example, Southern Germany exhibits higher water retaining capacities and also higher correlation with three month precipitation as compared to Eastern Germany. However, a substantial**

share of the variability of soil moisture is not explained by precipitation (see also comment above). One advantage of using soil moisture in such a study is that we can restrict the coefficients to be the same at all locations, whilst assuming that the water retaining is not the same everywhere. We added this point to the discussion of the results (page 11, line 32 - page 12, line 4 in the marked-up manuscript).

*[17] Page 12, line 8: 409 % with respect to what?*
In this context, 409.1 % refers to the relative difference of the adjusted R-square derived least squared dummy variable regression with respect to the one derived by the demeaning framework. In other words, it is the explanatory power added if the average yield of each county is explicitly taken into account in comparison to the models that only use soil moisture and weather variations as explanatory variables. We clarified this in the revised text on page 13, lines 1 - 16.

*[18] Page 13, line 15: Predictive power or explanatory power? This study does not assess the out-of-sample prediction performance; therefore, I would characterize adj. R2 as explanatory power. Can these models be used for silage maize yield seasonal forecasting? Out-of-sample validation would be necessary to determine the predictive power, especially due to the fact that relatively short time series are used to construct the regression model. Can you comment on that?*
We agree with the reviewer that we only assessed the explanatory power of the model in this study and adjusted the test accordingly (page 14, line 4 in the marked-up manuscript). It is planned to use the models of these studies for seasonal forecasting and climate projections in follow-up studies. Part two: In this study here, we used BIC to benchmark the model configurations, because it penalizes model over-fitting which hampers the out-of-samples predictability. However, follow up studies should perform an out-of-sample validation to determine the predictive power.

*[19] Page 16, lines 5-11: What could be the biophysical meaning behind the second and third terms being significant?*
We want to highlight here that it is very challenging to relate biophysical process to the coefficients of the statistical model used in this study. An approach using a processed-based crop model would be required to uncover these relationships. The main objective of considering non-linearities is to represent large and non marginal changes in the weather system (due to climate change) that are outside of the optimal plant growth conditions (changed in Section 1 & 4.1 (page 2, lines 18 - 23 & page 10, lines 9 - 14 in the marked-up manuscript). Therefore, non-linear dose-response functions are implemented. Furthermore, in this setting the terms of the polynomials are not orthogonal to each other. Thus, the non-significance of a term cannot be equated with a test for non-linearity. On the basis of the study design, the common significance of all polynomials rather implies a causal effect.

**References**

[revised manuscript text omitted]

C1: severe drought    C3: abnormally dry    C5: abundantly wet
C2: moderate drought    C4: abnormally wet    C6: severely wet

[revised manuscript text omitted]